# Fine-scale patterns of SARS-CoV-2 spread from identical pathogen sequences

Cécile Tran-Kiem[1 ✉], Miguel I. Paredes[1,2], Amanda C. Perofsky[3,4], Lauren A. Frisbie[5], Hong Xie[6], Kevin Kong[6], Amelia Weixler[6], Alexander L. Greninger[1,6], Pavitra Roychoudhury[1,6], JohnAric M. Peterson[5], Andrew Delgado[5], Holly Halstead[5], Drew MacKellar[5], Philip Dykema[5], Luis Gamboa[3], Chris D. Frazar[7], Erica Ryke[7], Jeremy Stone[3], David Reinhart[3], Lea Starita[3,7], Allison Thibodeau[5], Cory Yun[5], Frank Aragona[5], Allison Black[5], Cécile Viboud[4] & Trevor Bedford[1,8]

Pathogen genomics can provide insights into underlying infectious disease transmission patterns[1,2], but new methods are needed to handle modern large-scale pathogen genome datasets and realize this full potential[3–5]. In particular, genetically proximal viruses should be highly informative about transmission events as genetic proximity indicates epidemiological linkage. Here we use pairs of identical sequences to characterize fine-scale transmission patterns using 114,298 SARS-CoV-2 genomes collected through Washington State (USA) genomic sentinel surveillance with associated age and residence location information between March 2021 and December 2022. This corresponds to 59,660 sequences with another identical sequence in the dataset. We find that the location of pairs of identical sequences is highly consistent with expectations from mobility and social contact data. Outliers in the relationship between genetic and mobility data can be explained by SARS-CoV-2 transmission between postcodes with male prisons, consistent with transmission between prison facilities. We find that transmission patterns between age groups vary across spatial scales. Finally, we use the timing of sequence collection to understand the age groups driving transmission. Overall, this study improves our ability to use large pathogen genome datasets to understand the determinants of infectious disease spread.

Pathogen transmission is impacted by a multiplicity of factors associated with individual, population and environmental characteristics. As exposure and transmission are not directly observed, evaluating the contribution of these different factors to epidemic dynamics generally proves difficult. However, to anticipate the burden associated with epidemics and guide control policies, it is pivotal to understand how these different elements shape transmission risk.

Sequence data can provide insights into the proximity of individuals in a transmission chain. Phylogeographical approaches have helped to characterize how pathogens spread between different geographical regions[6,7] and demographic groups[8]. However, these methods currently face multiple limitations. First, they do not scale well past a few hundred or few thousand sequences owing to difficulties in scaling phylogenetic tree inference. Second, conclusions can be highly biased when sequencing is uneven[9]. Thus, we need new methods to analyse large pathogen genome datasets, such as those produced during the COVID-19 pandemic, which number in the millions of genomes[10].

As mutations accrue over the course of transmission events, we expect epidemiologically related individuals to be infected by pathogens that are genetically similar. Genetic-distance cut-offs have been used to distinguish plausibly-linked infections from infections resulting from distinct introductions within densely sampled outbreaks such as healthcare facilities or nursing homes[11,12]. Here we build from this expectation to characterize transmission patterns at the population level.

We introduce a statistical framework describing the relative risk (RR) of observing genetically proximal sequences in specific subgroups of the population. Our metric of association accounts for heterogeneity in sequencing effort between sampled locations and does not require building a phylogenetic tree, therefore making this approach directly scalable to large pathogen genomic datasets. We use this framework to investigate the spatial and social drivers of severe acute respiratory syndrome coronavirus 2 (SARS-CoV-2) transmission in Washington state (WA) by analysing 114,298 sequences (with associated age and home location information) collected through genomic sentinel surveillance in WA between March 2021 and December 2022.

## Spatio-temporal signal in identical sequences

As mutations accrue over time in pathogen sequences, individuals who are close together within a transmission chain are expected to be infected by genetically proximal viruses (Fig. 1a). For example, we expect that 64% of individuals infected with SARS-CoV-2 are infected

[1]Vaccine and Infectious Diseases Division, Fred Hutchinson Cancer Center, Seattle, WA, USA. [2]Department of Epidemiology, University of Washington, Seattle, WA, USA. [3]Brotman Baty Institute, University of Washington, Seattle, WA, USA. [4]Fogarty International Center, National Institutes of Health, Bethesda, MD, USA. [5]Washington State Department of Health, Shoreline, WA, USA. [6]Department of Laboratory Medicine and Pathology, University of Washington, Seattle, WA, USA. [7]Department of Genome Sciences, University of Washington, Seattle, WA, USA. [8]Howard Hughes Medical Institute, Seattle, WA, USA. ✉e-mail: ctrankie@fredhutch.org

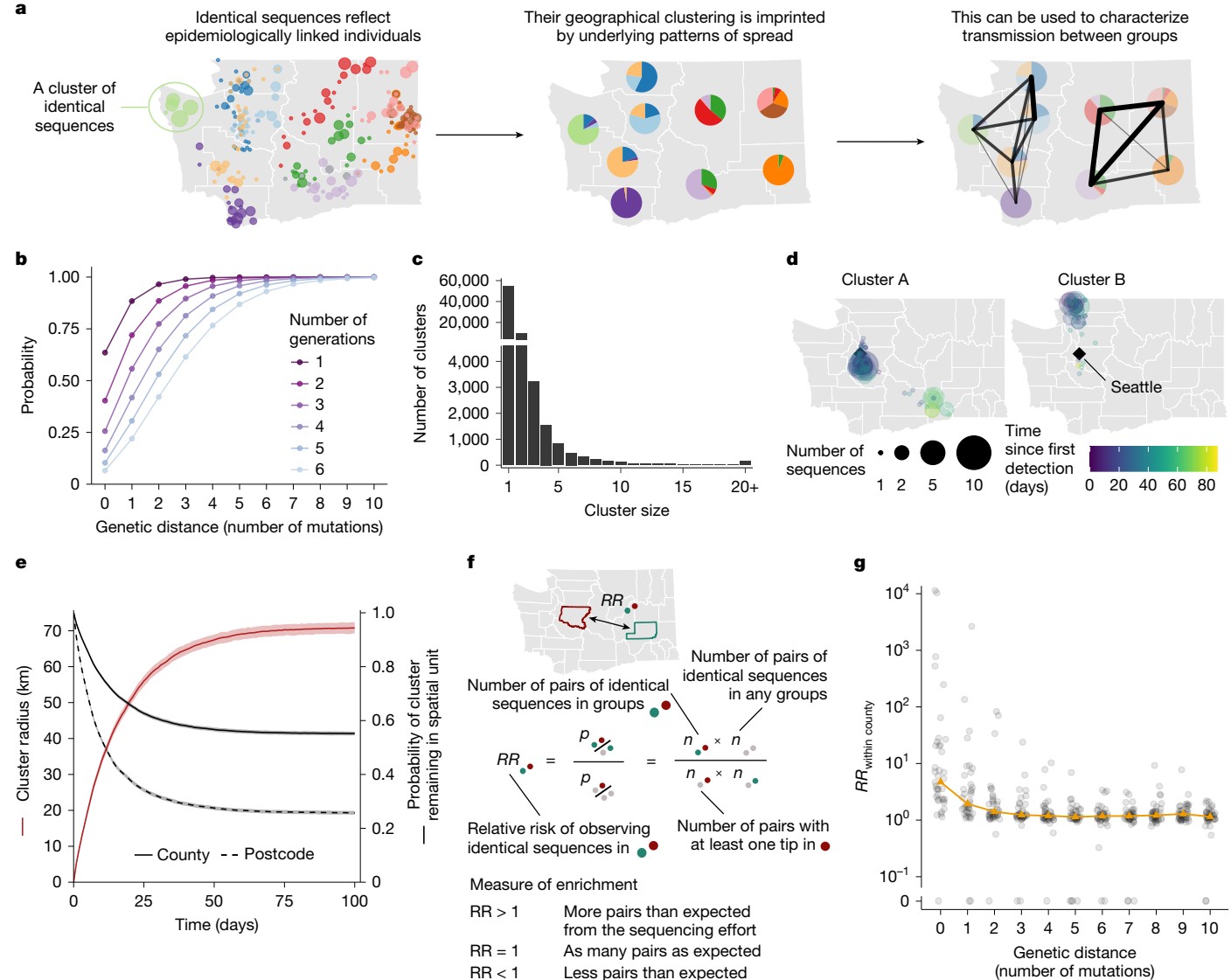

**Fig. 1 | Temporal and spatial signature of spread in clusters of identical SARS-CoV-2 sequences. a**, Clustering of identical pathogen sequences across population groups reflects underlying disease transmission patterns at the population level and can be used to characterize spread patterns between groups. Each colour represents a different cluster of identical sequences. **b**, Probability for two individuals separated by a fixed number of transmission generations of being infected by viruses at a given genetic distance assuming a Poisson process for the occurrence of substitutions (at a rate $\mu = 8.98 \times 10^{-2}$ substitutions per day) and gamma-distributed generation time (mean, 5.9 days; s.d., 4.8 days). **c**, Size distribution of clusters of identical sequences in the WA dataset. Clusters of size 1 correspond to singletons and are therefore not included in the RR computations. **d**, Spatiotemporal dynamics of sequence collection in two large clusters of identical sequences. The black diamonds indicate the location of Seattle, the largest city in WA. **e**, Radius of clusters of identical sequences (red line) and probability for all sequences within a cluster of identical sequences of remaining in the same spatial units (black lines) as a function of time since first sequence collection. In **e**, the cluster radius is computed as the mean spatial expansion of clusters of identical sequences. **f**, Definition of the RR of observing pairs of sequences in two subgroups as a measure of enrichment. **g**, RR of observing pairs of sequences within the same county as a function of the genetic distance separating them. The grey points correspond to values for individual counties. The orange triangles correspond to the median across counties. For **a**, **d** and **f**, maps were generated using shapefiles from the US Census Bureau[44].

by a virus with the same consensus genome as their infector (Fig. 1b). Identical sequences should therefore be highly informative about SARS-CoV-2 transmission events as they are preferentially collected from the most epidemiologically linked individuals. Thus, their geographical clustering should be informative about spatial patterns of transmission. Here we leverage the clustering of identical sequences between groups to characterize transmission at the population level (Fig. 1a). In WA, we identified 17,231 clusters of identical sequences excluding singletons, corresponding to 59,660 sequences (Fig. 1c). In some large clusters of identical sequences, we observed local spread before wider geographical expansion (Fig. 1d). Using postcodes and collection dates, we estimated cluster radius in kilometres. Across

clusters, we find that the spatial expansion of clusters increases over time (Fig. 1e) and is significantly lower than expected at random (Supplementary Fig. 1). The probability for a cluster to remain within the county and zip code where it was first identified decreases over time. These probabilities are significantly higher than expected at random (Supplementary Fig. 1). This confirms that clusters of identical sequences contain a strong spatial and temporal signature of spread.

## RR framework

To quantify the association between subgroups of the population (such as geographical units or age groups) from genetic data, we introduce a

measure of RR describing how the number of pairs of sequences separated by a fixed genetic distance observed in two subgroups differs from what we expect from the sequencing effort (Fig. 1f and Methods). This RR can be interpreted as a measure of enrichment describing how the number of pairs shared by these two subgroups differs from what we expect from the overall number of pairs observed in these two subgroups.

Figure 1g shows the relationship between the RR of observing sequences within the same county and the genetic distance between pairs. Among all counties, the median RR of observing identical sequences within the same county is equal to 4.7 (interquartile range = 2.4–21.2) across the time period. When considering a greater genetic distance between pairs, this signal decreases to plateau at 1. This confirms that the location of genetically close sequences (less than a couple of mutations away) and especially identical sequences is informative about local spread patterns, wherein infected individuals transmit more often within their home county.

We observe this trend across variants and periods (Extended Data Fig. 1). The magnitude of the absolute RR along with the speed at which it decays as a function of genetic distance vary. For example, during the period in which the prevalence of the Omicron SARS-CoV-2 variant rises and the Delta SARS-CoV-2 variant declines, the RR of observing identical sequences in the same county is higher among Delta than Omicron sequences. This can be explained by differences in transmission intensity: a higher transmission rate results in larger clusters of identical sequences[13] that will tend to be more geographically widespread (Extended Data Fig. 1c). The spatial signal from genetically close sequences is therefore weaker in periods characterized by a higher transmission intensity. Other factors, such as changes in mixing and travels patterns, can also impact the magnitude of the RR.

Sampling biases can considerably impact the results of phylogeographical inference[9]. Here, although the proportion of pairs of identical sequences observed in a county is highly correlated with the number of sequences observed in this county, we find that the RR is no longer correlated with sequencing effort (Extended Data Fig. 2). Using a simulation approach, we show that our RR metric captures the migration probability between population subgroups, including when sequencing effort is heterogeneous (Extended Data Fig. 3 and Supplementary Table 1). This contrasts with migration rates obtained from a discrete trait analysis (DTA)[14] that are poorly correlated with true migration rates when sequencing effort differs between regions (Extended Data Fig. 3 and Supplementary Table 1). These DTA results are obtained by inputting the exact simulated transmission tree. In practice, inferring the underlying tree will decrease accuracy due to phylogenetic uncertainty so that these DTA estimates represent an upper bound of the DTA's potential performance. If we compare DTA accuracy between input phylogeny and phylogeny estimated from sequence data, we find that Pearson correlation between true migration rates and estimated migration rates changes from 0.54 to 0.10 for unbiased sampling and changes from −0.22 to 0.15 for biased sampling (Supplementary Table 1). Running the phylogenetic DTA analysis on simulated data with 1,745 sequences requires 1 day when using the empirical tree and 24 days when jointly inferring the tree and the migration rates (Methods). Running our RR analysis on the same sequence dataset takes 33 s. This result demonstrates that the RR framework constitutes an appropriate approach to study the determinants of SARS-CoV-2 transmission by explicitly accounting for sequencing effort and uneven sequencing between population subgroups.

## Patterns of SARS-CoV-2 spread between WA counties

We examined the geographical spread by analysing patterns of occurrence of identical sequences in WA counties (Fig. 2). The matrix of pairwise RRs between counties (Supplementary Fig. 2) is characterized by a strong diagonal, which is consistent with within-county transmission. To better understand the spatial patterns of SARS-CoV-2 spread between counties, we display these RRs on choropleth maps indicating the RR for different focal counties (Fig. 2a and Supplementary Fig. 3). These maps suggest that identical sequences have a higher risk of falling within counties that are geographically nearby. Across all pairs of counties, we find a geographical gradient in the RR of identical sequences, whereby the risk is highest within the same county, intermediate between adjacent counties and lowest between non-adjacent counties (Fig. 2b). The risk of observing identical sequences between counties also decays as a function of geographical distance (Fig. 2c) and is no longer significant at distances greater than 177 km (95% confidence interval (CI) = 137–241).

To assess whether global spatial structure is maintained, we implemented a multidimensional scaling (MDS) algorithm by defining a similarity metric based on the RR of observing identical sequences between counties. MDS enables us to display the relatedness of observations based on a distance matrix. This MDS ordination shows county relationships that recapitulate the Western (WWA) and Eastern (EWA) WA regions, two regions that are separated by the Cascades mountain range (Fig. 2d). Within EWA and WWA, we find a strong signal for local spread, with identical sequences having a higher risk of being observed between adjacent than between non-adjacent counties (Fig. 2e). Across the EWA–WWA border, we no longer find that identical sequences have an increased risk of being observed in adjacent counties. Results are similar when analysing pairs of identical sequences at the postcode level (Supplementary Table 2). This lack of association is not affected by the low number of pairs of adjacent counties across the EWA–WWA border (Supplementary Fig. 4). This illustrates how heterogeneous physical landscape features can impact and distort patterns of disease spread and genetic diversity[15–18]. We also find that the association between the RR of observing identical sequences in two counties is significant at greater distance within EWA than within WWA (Fig. 2f). We do not find any association with distance across the EWA–WWA border, although this might be explained by the lack of counties with low distances across the EWA–WWA border.

Finally, we find that, across epidemic waves, pairs of identical sequences observed on both sides of the Cascades are consistently observed first in WWA (Fig. 2g and Supplementary Fig. 5). As testing behaviour and access to healthcare can be influenced by county demographic characteristics and how rural or urban a county is, we examined how this trend varied when using symptom onset dates instead of sequence collection dates, which provides similar trends (Fig. 2g). Despite the existence of negative serial interval for SARS-CoV-2[19], this analysis provides direct insights into the typical transmission direction between groups as the proportion of SARS-CoV-2 transmission pairs with positive serial intervals in greater than 50% (ref. 20) (Supplementary Note 1). This asymmetry suggests that identical sequence clusters tend to percolate from WWA to EWA more so than the reverse, indicating that transmission generally flows from WWA to EWA. This trend is similar to the one reported in phylogeographical analyses of the first COVID-19 wave in WA that concluded that more introductions occurred from WWA to EWA than from EWA to WWA[21].

## Relationship with human mobility

We next examined the extent to which spatial transmission patterns inferred from identical sequences can be explained by human mobility indicators. To compute the RR of movement between two counties or regions, we used aggregated mobile phone location data obtained from the SafeGraph 'Weekly Patterns' dataset and pre-pandemic commuting data from the US Census Bureau[22] (Methods). Despite commuting data being collected before the pandemic and mobile phone location data being collected during our study period, we found that these two mobility data sources are highly correlated (Supplementary Fig. 6). We assessed how the RR of observing identical

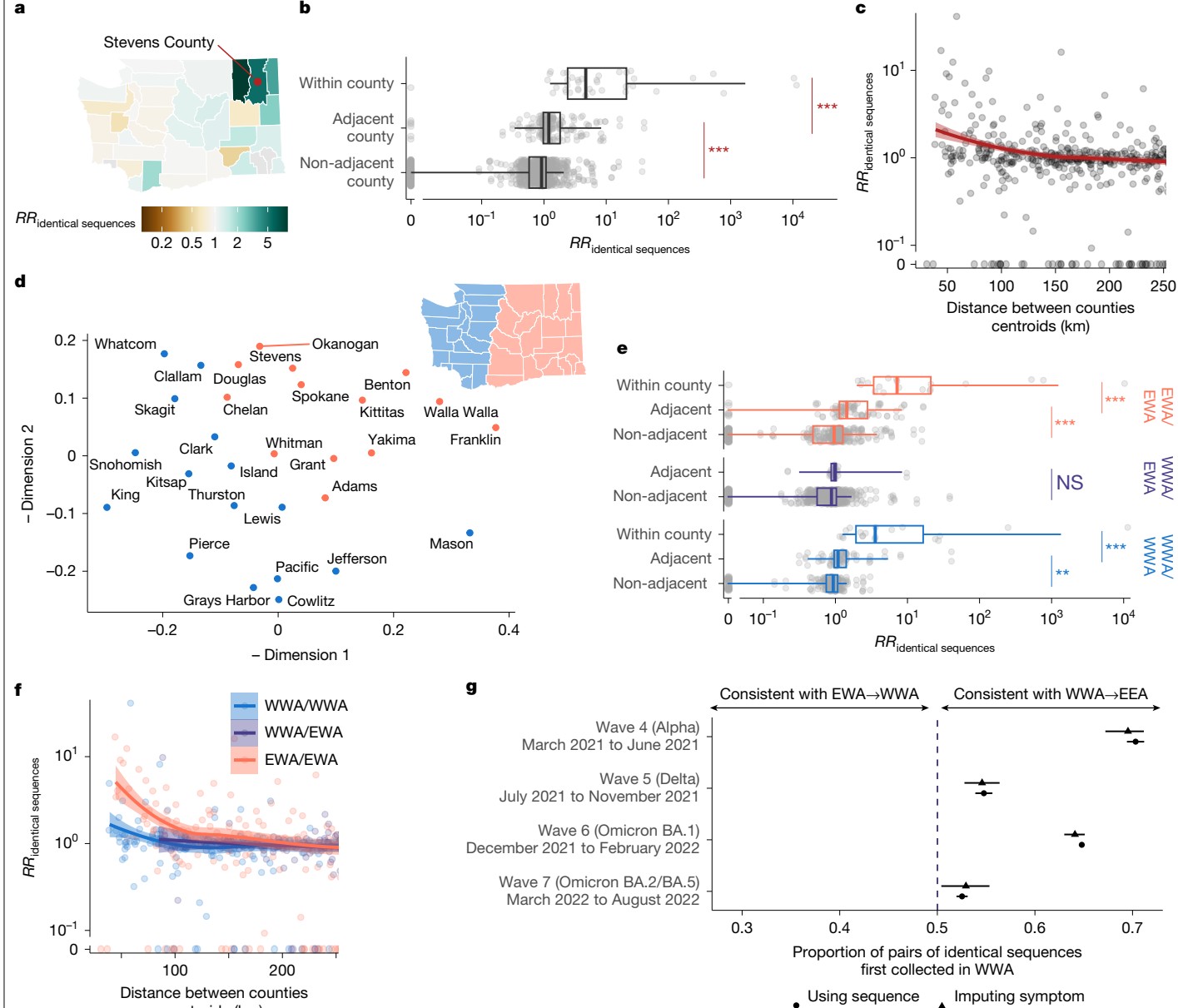

**Fig. 2 | Identical sequences reveal patterns of spread between WA counties.**
**a**, Illustration of the pairwise RR of observing identical sequences between counties, using sequences shared between Stevens County (red point) and other counties in WA as an example. Similar maps for the other counties are depicted in Supplementary Fig. 3. **b**, RR of observing pairs of identical sequences by counties' adjacency status. **c**, RR of observing pairs of identical sequences as a function of the geographical distance between counties' centroids. **d**, Similarity between WA counties obtained from MDS based on the RR of observing pairs of identical sequences in two counties. Counties are coloured by east–west region membership. **e**, RR of observing pairs of identical sequences by counties' adjacency status stratified by counties east–west region membership. **f**, RR of observing pairs of identical sequences as a function of the

geographical distance between counties' centroids stratified by counties east–west region membership. **g**, Proportion of pairs of identical sequences observed in EWA and WWA that were first observed in WWA. In **c** and **f**, the lines correspond to LOESS curves on the logarithmic scale. In **b** and **e**, $P$ values calculated using Wilcoxon tests are as follows: ***$P < 0.0001$, **$P < 0.001$, *$P < 0.05$; NS, $P \geq 0.05$. In **b**, $W_{within,adjacent} = 6,195$ ($P = 3.7 \times 10^{-12}$) and $W_{adjacent,non-adjacent} = 65,542$ ($P < 2.2 \times 10^{-16}$). In **e**, for within EWA, $W_{within,adjacent} = 120.5$ ($P = 6.7 \times 10^{-6}$) and $W_{adjacent,non-adjacent} = 4,555.5$ ($P = 4.0 \times 10^{-6}$). For within WWA, $W_{within,adjacent} = 95$ ($P = 9.9 \times 10^{-7}$) and $W_{adjacent,non-adjacent} = 3626$ ($P = 1.1 \times 10^{-4}$). For between EWA and WWA, $W = 2,719$ ($P = 0.17$). For **a** and **d**, maps were generated using shapefiles from the US Census Bureau[44].

sequences in two counties relates to the RR of movement (Fig. 3a, Extended Data Fig. 4 and Extended Data Table 1) by implementing a generalized additive model (GAM) that includes a single predictor of smoothed RR of movement between two counties as a covariate. We used a GAM rather than linear regression as we expect the functional form of the relationship to be nonlinear (Supplementary Fig. 7). This nonlinearity can be explained by the indirect mapping between transmission events and identical sequences that encompass

both direct transmission pairs and pairs of individuals a couple generations apart.

When comparing RRs at the county level, we found that 60% of the variance in identical sequence data is explained by between-county flows derived from the mobile phone data (Fig. 3a and Extended Data Table 1). For a subset of counties, the number of pairs of identical sequences or the number of trips reported in the mobility dataset is low. For these low counts, we expect RRs to be more noisy.

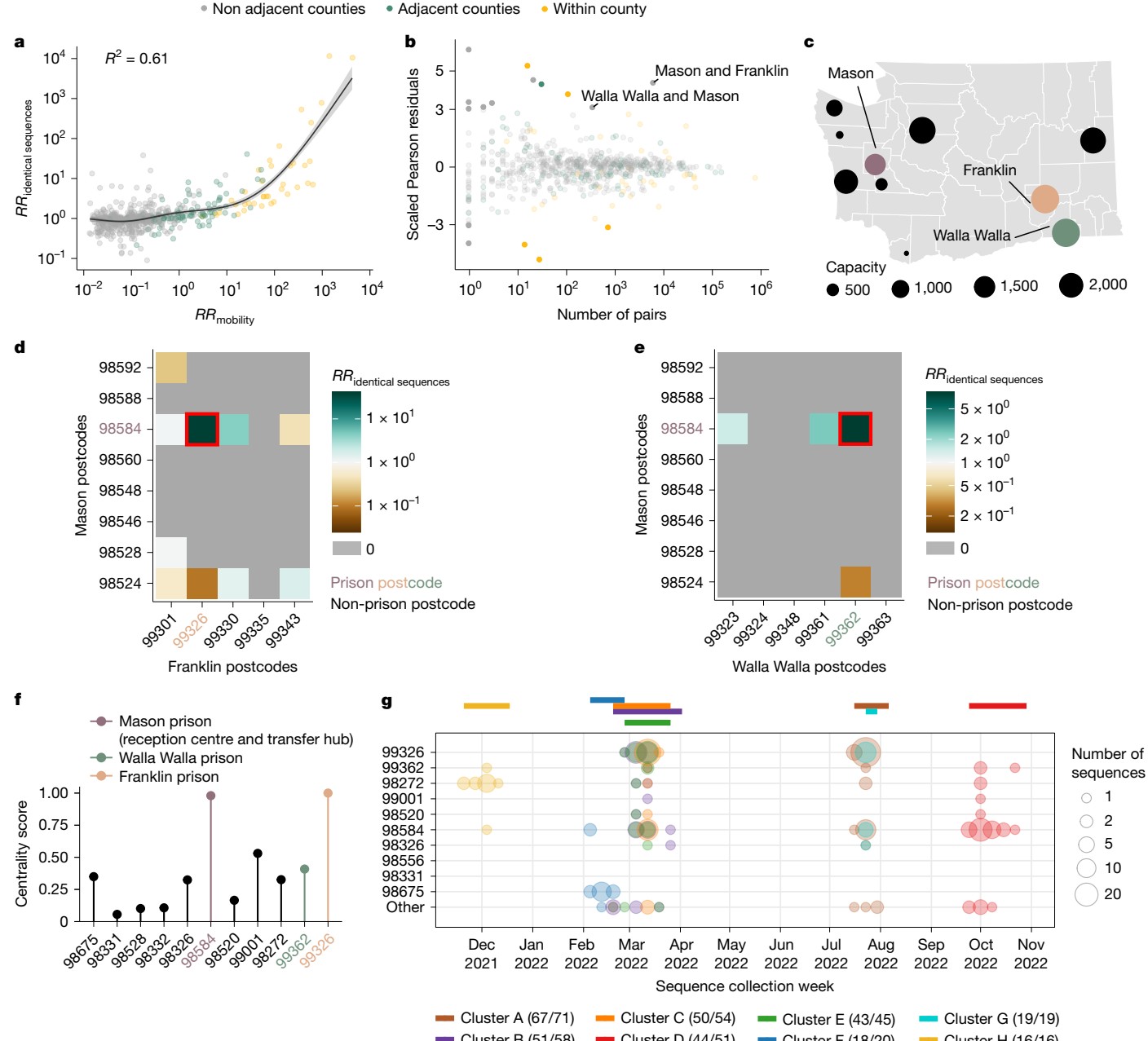

**Fig. 3 | Comparison of the location of identical sequences with expectations from mobility data reveals spread between WA male prison postcodes.** **a**, Relationship between the RR of observing identical sequences in two counties and the RR of movement between these counties as obtained from mobile phone mobility data. The trend line corresponds to the predicted RR of observing identical sequences in two regions from a GAM. The $R^2$ indicates the variance explained by the GAM. **b**, Scaled Pearson residuals of the GAM plotted in **a** as a function of the number of pairs of identical sequences observed in pairs of counties. **c**, Map of male state prisons in WA. Mason, Walla Walla and Franklin male prisons are coloured. **d**, RR of observing identical sequence between Mason and Franklin County's postcodes. **e**, RR of observing identical sequence between Mason and Walla Walla County's postcodes. **f**, Centrality score (eigenvector centrality) for each postcode that is the home of a male state prison. **g**, Week of sequence collection within eight large clusters of identical sequences identified in postcodes with WA male state prisons. In **g**, the top coloured segments indicate the period during which each cluster was identified. For **c**, maps were generated using shapefiles from the US Census Bureau[44].

To remove potential noise associated with these lower counts, we repeated this analysis at a larger spatial scale. Aggregating pairs at the regional level (9 regions in WA for 39 counties; Supplementary Fig. 8) increases the variance explained to 81% (Supplementary Fig. 9). We also find that pre-pandemic workflow data are highly informative of the spatial distribution of pairs of identical sequences with a similar strength of relationship to that observed for mobile phone mobility data (Extended Data Fig. 4, Supplementary Fig. 9 and Extended Data Table 1).

Non-pharmaceutical interventions along with behavioural changes have impacted human mobility patterns throughout the COVID-19 pandemic. We find that mobility data explain a high percentage of variance in the RR of observing identical sequences between WA regions across individual epidemic waves (Supplementary Fig. 10) but not to a greater extent than over the entire study period. This can probably be explained by the high stability of the structure of the mobility network between WA counties across epidemic waves (Supplementary Fig. 11). This suggests that analysing COVID-19 waves separately tends

to introduce noise rather than increase the spatial resolution, consistent with a former analysis that concluded that there was a high stability of between-county mobility patterns during the beginning of the pandemic in the United States[23].

Among counties located across the EWA–WWA border, the risk of movement across the border is lower than the risk of movement within the same region (Supplementary Fig. 12). This shows that human mobility is highly predictive of the location of pairs of identical sequences and explains some of the spatial patterns reported in Fig. 2.

### Association between outliers and male prisons

We identified unexpected patterns of transmission between counties from outliers in the relationship between mobility and genetic data (Fig. 3a). We define outliers as pairs of counties for which the absolute value of the scaled Pearson residual from the GAM is greater than 3. As we expect RRs computed from a low number of pairs of identical sequences to be noisier, we focused on pairs of counties between which at least 100 pairs of identical sequences are observed. We found unexpected patterns of SARS-CoV-2 spread between two non-adjacent pairs of counties, Franklin–Mason and Walla Walla–Mason counties, with more pairs of identical sequences observed than expected from mobility data (Fig. 3b). The association between Franklin and Mason (RR of 13.4, 95% CI = 11.4–16.4) and Walla Walla and Mason (RR of 5.9, 95% CI = 4.0–8.3) is particularly surprising given that they are non-adjacent counties located on different sides of the Cascades. As no demographic or geographical factors provide a straightforward explanation for such an association, we hypothesized that such a pattern might arise from SARS-CoV-2 spread on a dissemination network that differs from the general community. We identified that these three counties are the home of male state correction centres (Fig. 3c). We also found that identical sequences have a higher risk of being observed within Lincoln County and a lower risk of being observed within Pacific County than expected from cellphone mobility data, without identifying any demographic factor explaining these associations.

To investigate whether the unexpected pattern of association between Franklin and Mason, and Walla Walla and Mason counties can be explained by transmission within the prison network, we looked at patterns of association between Franklin and Mason and Walla Walla and Mason postcodes (Fig. 3d,e). For most of these pairs of postcodes, we do not observe any pair of identical sequences throughout the study period. Notably, for each pair of counties, the genetic signal can be explained by a high RR of observing identical sequences between two postcodes, which correspond to the postcodes that are the home to the male correction centres that we identified. The greater number of pairs of identical sequences observed between Mason and Franklin and Mason and Walla Walla counties than expected from mobile-phone-derived mobility data can therefore be explained by a large number of pairs of identical sequences in specific postcodes with male correction centres.

We also investigated patterns of occurrence of pairs of identical sequences between the two counties (Mason and Pierce) that are the home to female prisons. At the county level, identical sequences do not have an increased risk of occurring between Mason and Pierce counties (RR of 0.59, 95% CI = 0.49–0.67). However, at the postcode level, we found that the RR of observing identical sequences is highest between the two postcodes with female prisons (Supplementary Fig. 13). This shows how our framework enables exploration of patterns of spread at different spatial scales: we do not find any signal at the county level, probably because Mason and Pierce are adjacent counties, but we can identify association at the postcode level.

It is interesting that the pairs of outliers that we identified systematically involved Mason County (Fig. 3b), which is the home of only the sixth (out of ten) most populated male prison in the state (Fig. 3c). The prison in Mason County (Washington Corrections Center) has a particular role in the WA prison network as it serves both as a reception centre for anyone entering the WA prison system and as a transfer hub[24]. To understand whether the prison network structure can explain patterns of SARS-CoV-2 transmission, we conducted a centrality analysis. To do so, we analysed the network of postcodes with WA male prisons and we defined the weight of each edge by the RR of observing identical sequences between these two postcodes. We found that the two nodes with the highest eigenvector centrality scores are the postcodes that are the home of Washington Corrections Center (Fig. 3f) and of the Franklin County prison (most populated prison). This shows that patterns of occurrence of identical sequences in WA are imprinted by the structure of the prison network.

Finally, we investigated whether large clusters of identical sequences are shared between postcodes with male state prisons, which we define as clusters with more than 15 sequences in male state prisons postcodes. Figure 3g depicts the timing of the large clusters that we identified. Notably, the largest cluster (cluster A) includes 71 sequences collected between 18 July and 31 July 2022, 67 of which came from postcodes with male state prisons. The second largest cluster (cluster B) is composed of 58 sequences collected between 21 February and 29 March 2023, among which 51 came from 7 different prison postcodes. Notably, the postcode of Washington Corrections Center is the only one in which all of these eight clusters were observed.

Populations who are incarcerated have been particularly affected by the COVID-19 pandemic[25,26]. To mitigate the impact of the pandemic in these congregate settings, various interventions have been implemented. In WA, for example, testing followed by quarantine protocols were carried out in Washington Corrections Center after admission and before any transfer. Active screening of staff was also implemented throughout the pandemic. However, individuals incarcerated who were diagnosed with COVID-19 at times had to be transferred from Washington Corrections Center to other WA prisons due to the finite capacity of the reception centre. With vaccine mandates, staff also had to be relocated to cope with the departure of other employees. Our results reveal multiple SARS-CoV-2 introductions between WA prisons, that could be explained by the movements of both individuals incarcerated and staff.

This analysis showcases how identical sequences can help to identify under-recognised viral dissemination networks that differ from transmission pathways in the general community. The counties that we identified as outliers in the relationship between genetic and mobility data have a particularly high ratio between the prison population size and the county population size (between around 2% and 4%; Supplementary Table 3). This probably explains why we were able to detect this signal at the county level but had to investigate transmission at the postcode level to study transmission between other prisons.

### Age transmission patterns vary across spatial scales

Spatial and social factors (such as age) are key determinants of the spread of respiratory infections such as SARS-CoV-2 and influenza[27–30]. We expect movement patterns to differ between age groups (such as children, adults and older people), which can impact patterns of disease transmission[31–33]. However, there has been limited empirical evidence of this phenomenon and data sources that can be leveraged to characterize this interaction are critically needed. Here, we show that we can combine pathogen sequence information with detailed metadata to investigate how age mixing patterns vary across spatial scales.

We first examined whether we could recover the expected age-mixing signature from the sequence data before delving into the interaction between age and space. We found that the age groups in which identical sequences are observed are consistent with assortative mixing patterns and mixing between generations (Extended Data Fig. 5). Comparing this with expectations from synthetic social contact data for WA[34], we found that the signal obtained from identical sequences is highly

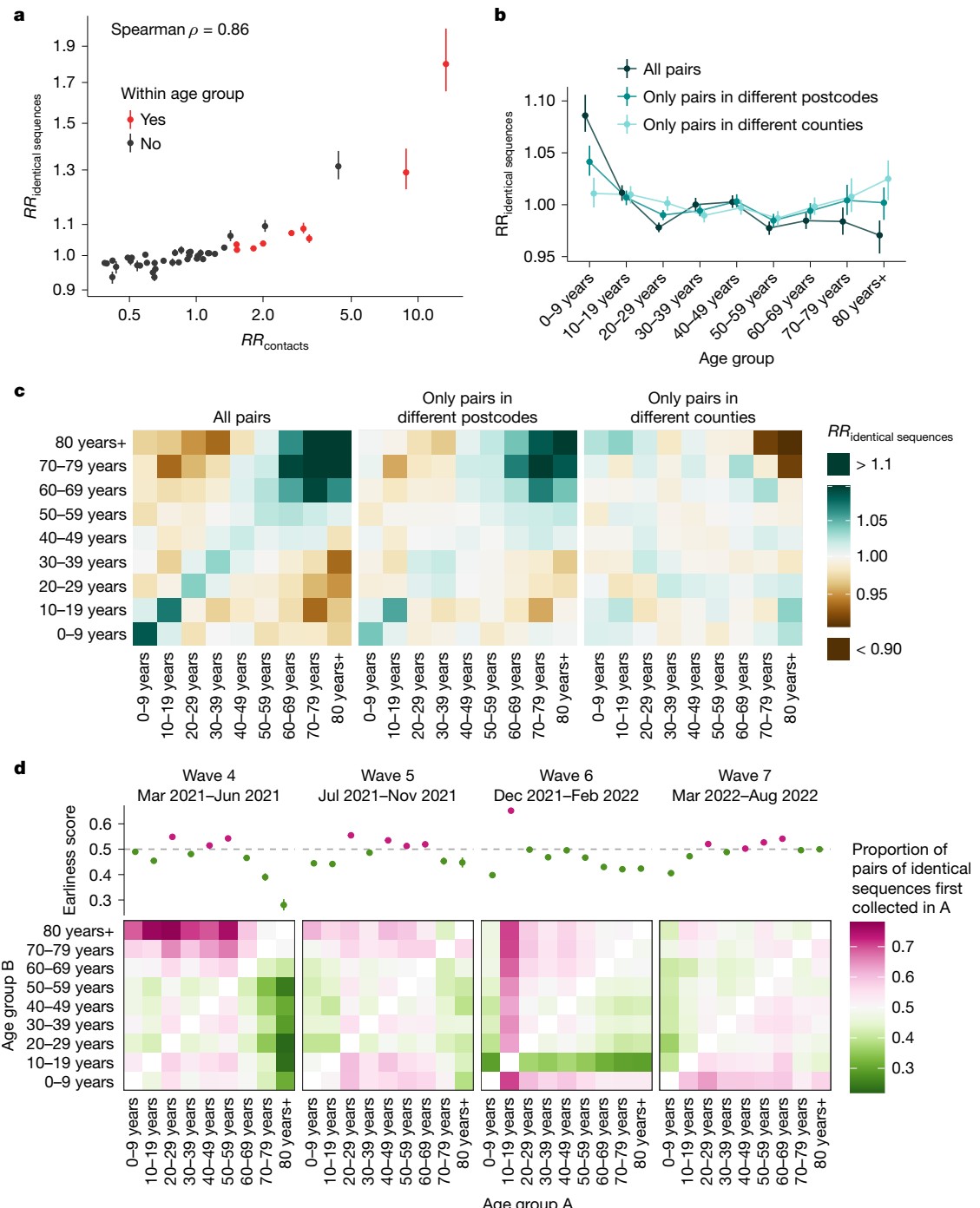

**Fig. 4 | Patterns of SARS-CoV-2 transmission between age groups in WA.** **a**, RR of observing pairs of identical sequences in two age groups as a function of the RR of contact between these age groups. **b**, Impact of the spatial scale on the RR of observing pairs of identical sequences in the 0–9 year and other age groups. We display similar plots for the other age groups in Extended Data Fig. 7. **c**, RR of observing identical sequences between two age groups across all pairs of sequences, only pairs in different postcodes and only pairs in different counties. **d**, Proportion of pairs of identical sequences observed in age groups A and B that were first collected in age group A across different epidemic waves (heat maps). The dot plots depict the earliness scores of age group A across epidemic waves. In **a** and **b**, the vertical segments correspond to the 95% subsampling CIs. In **d**, the vertical segments correspond to the 95% binomial CIs. In **d**, the heat maps represent symmetric matrices $P = (p_{i,j})$ characterized by $p_{i,j} + p_{j,i} = 1$.

correlated with that expected from age-mixing matrices (Fig. 4a; GAM: 90% of variance explained; Spearman $\rho$ = 0.86, $P < 10^{-16}$). The signal for SARS-CoV-2 transmission between generations (such as between the 0–9 year and 30–39 year age groups) fades out when considering pairs of sequences separated by a greater genetic distance (Extended Data Fig. 6). As sequences at a greater genetic distance come from individuals who are further apart within a transmission chain (Fig. 1a), fine-scale

patterns of spread might indeed not be apparent from sequences at more than a couple of mutations away. This emphasizes the value of analysing identical pathogen sequences to characterize subtle patterns of pathogen spread and population mixing, especially when population subgroups are very mixed.

Next, we compared the RR of observing identical sequences between two age groups by looking at either all pairs of sequences or only pairs

of sequences from individuals living in different counties or postcodes. We found that the spatial scale modulates patterns of disease transmission between age groups (Fig. 4b,c and Extended Data Fig. 7). We found that pairs of identical sequences coming from the same county and postcode are enriched in same-age pairs. This enrichment is particularly important in older groups. For example, the RR of observing identical sequences in individuals aged 80 years and older drops from 1.80 (95% CI = 1.65–2.01) when considering all pairs to 0.79 (95% CI = 0.71–0.85) when considering pairs coming from individuals living in different counties (Extended Data Fig. 7). This shows that transmission to and from older age groups tends to occur close to their home location and suggests that older individuals' typical radius of movement is smaller than that of other age groups. Only considering pairs of sequences in 0–9 year olds coming from different spatial units largely decreases the signal for SARS-CoV-2 transmission between children and adults aged 30–49 years (Fig. 4c). This is expected given that we anticipate that most of these contacts occur within the household[35]. Overall, we find that looking at patterns of occurrence of identical sequences at a greater geographical scale largely distorts the contact structure. For example, the location of identical sequences suggests that transmission to and from older individuals outside of their home counties tends to occur with younger age groups, including younger children (such as grandchildren).

Mixing patterns between age groups have been extensively studied[36,37]. A social contact survey performed in southern China reported that older individuals' contacts occurred closer to their homes compared with younger individuals' contacts[38]. However, overall, there has been limited evidence to quantify the spatial distribution of these contacts. Spatial mixing is generally measured from aggregated mobility data sources that generally do not provide demographic information such as age. As spatial and age mixing are reconstructed from different data sources, understanding their interplay has been difficult. Here we show that we can directly leverage pathogen genome data with linked age and spatial information to understand where age-specific transmission is occurring. This suggests that the wider availability of sequencing data might provide an opportunity to directly infer how population groups interact in a way that is relevant for pathogen spread, without the need to implement laborious contact surveys or collect mobility data.

## Timing of identical sequence collection

Finally, we used the timing of identical sequence collection to investigate the age groups driving SARS-CoV-2 transmission over the course of the pandemic in WA. Within pairs of identical sequences, we indeed expect age groups acting as sources to be consistently detected before groups acting as sinks (Supplementary Fig. 14). In Fig. 4d, we display for every age group combination and across epidemic waves the proportion of pairs of identical sequences first collected in a given age group. During the fourth and fifth pandemic wave in WA (mainly caused by the Alpha and Delta SARS-CoV-2 variants of concern, respectively), we found that pairs of identical sequences are consistently observed later in older groups even though the RR of observing identical sequences in older groups and younger groups is low (Extended Data Fig. 5). This could be consistent with younger age groups acting as source of infections for older individuals. During the fourth pandemic wave, sequences from individuals aged 20–29 years and 40–59 years are systematically observed before any other groups within pairs of identical sequences and likewise during the fifth pandemic wave, sequences from individuals aged 20–29 years and 40–69 years are observed earlier than other age groups. This could be consistent with these groups acting as sources of infection for the other age groups. During the sixth wave, sequences from individuals aged 10–19 years tend to be observed first within pairs of identical sequences, which suggests their role as sources for other ages groups and corresponds to the Omicron wave during

a time when schools had recently returned to in-person instruction. From March 2022, the contribution to transmission is more evenly distributed across age groups.

The role played by young children during the COVID-19 pandemic has been highly debated[30,39]. Here we find that, during the Alpha and Delta epidemics (waves 4 and 5), children aged 0–9 years could have acted as a source of SARS-CoV-2 infections for older individuals but not for younger adults. This pattern disappears during the Omicron epidemic (wave 6) in which pairs of identical sequences tend to be observed first in other age groups before being collected in young children aged 0–9 years. This could be explained by behavioural changes or by different immune profiles across age groups, resulting in different relative susceptibility to Omicron relative to Delta[40]. Overall, we do not find an indication that young school-age children act as major sources of SARS-CoV-2 transmission in the population, even after schools reopened.

Overall, our results highlight the porosity of SARS-CoV-2 transmission across age groups and suggest a role played by lower-risk groups in seeding infections in higher-risk groups. We come to similar conclusions when looking at the timing of symptom onset dates (Supplementary Fig. 15), which suggests that our conclusions are robust to differences in testing behaviours across age groups. Our conclusions are consistent with existing literature emphasizing the important role played by young adults and teenagers and the limited contribution of children and older individuals in driving SARS-CoV-2 spread[30,41]. Analysing the timing of identical sequence collection provides immediate insights into pathogen flow between population subgroups.

Here, we focus on understanding patterns of SARS-CoV-2 spread between age groups, but this approach can be applied to investigate the spread of fast-evolving pathogens between various demographic groups, such as occupational and ethnic groups, behavioural and risk groups. For example, we find that identical sequences collected within an age group tend to be enriched with same-vaccine status pairs during the Alpha, Delta and Omicron BA.2/BA.5 waves in WA (Supplementary Fig. 16). Social clustering of unvaccinated individuals or generally individuals with a different immune background can have important implications regarding the size and likelihood of infectious disease outbreaks[42,43]. This suggests that our approach has the potential to shed light on such a phenomenon and more generally on broad determinants of disease transmission.

## Caveats

Although our RR metric explicitly accounts for sampling intensity in locations in which pairs of identical sequences are collected, it cannot describe patterns of spread from non-sampled locations. In theory, unsampled locations could impact our assessment of local transmission patterns if two individuals with an identical sequence are both infected by someone outside the study area. However, in practice, we find that non-sampled locations have little impact on our RR computation (Extended Data Fig. 8). This suggests that background sequences are not required to evaluate SARS-CoV-2 transmission at the state level. Other epidemiological settings might nonetheless require more careful considerations, for example, in the hypothetical scenario in which a non-sampled location is responsible for the majority of the infections within the study area.

Compared with existing phylogeographical methods, our approach does not require including background sequences from outside locations, as non-sampled locations have little impact on the RR computation (Extended Data Fig. 8). Our approach could also overestimate RRs associated with transmission events that are over-represented in the sequencing data. For example, applying this analysis to sequences predominantly collected through household studies could overestimate the contribution of contacts within the household to the overall infection burden. In our case, patterns of occurrence of identical sequences in WA are potentially affected by intensive testing performed during

outbreaks within WA prisons during the pandemic. Part of the signal that we have detected might therefore come from a higher sequencing rate in prisons compared with in the general community. However, the very large clusters of identical sequences shared between multiple prison postcodes confirm that SARS-CoV-2 extensively spread within the prison network.

Our results suggest that the timing of sequence collection within clusters of identical sequences provides valuable information to understand transmission direction. Here, we report a simple quantity based on the proportion of pairs first observed in a group that summarizes the earliness of a group compared with another within pairs of identical sequences. Such an earliness metric might incompletely capture the transmission process as clusters can for example span more than two groups. We performed a sensitivity analysis in which we relied only on clusters of identical sequences observed in two groups, providing similar results (Supplementary Fig. 17). Overall, more work is warranted to robustly quantify transmission rates from the timing of identical sequences.

Finally, the magnitude of RRs is impacted by transmission intensity (Extended Data Fig. 1), so that values computed between two regions across different time periods cannot be directly compared. However, within a time period, the ranking of RRs is informative about patterns of transmission between groups. Overall, further work exploring how patterns of occurrence of identical sequences can be used to directly infer mixing rates between groups, while incorporating temporal changes in transmissibility would be particularly interesting.

## Applicability beyond SARS-CoV-2

Here we studied patterns of SARS-CoV-2 transmission between geographies and age groups in WA using a particularly rich sequence dataset, both given the amount of sequences available and the quality of the associated metadata. However, this work can readily be applied to other densely sampled pathogens.

The power of our method is determined by the number of pairs of identical sequences available, which will be impacted by transmission intensity (higher reproduction numbers will tend to result in larger clusters of identical sequences[13]) and the relative timescale at which substitution and transmission events occur[13]. Overall, this approach is well tailored to study densely sampled outbreaks. Compared with phylogeny-based methods of which the power comes from the number of unique haplotypes, our ability to characterize spread from identical sequences depends on the number of haplotypes with multiple observations. Whereas we expect diminishing returns when sequencing a greater proportion of cases using phylogeny-based methods, with a decreasing number of new haplotypes per additional sequence, the number of haplotypes with multiple occurrences will increase for each additional sequence included in the dataset (Supplementary Fig. 18). The number of population groups included in the analysis will also impact the amount of sequencing data required. In situations in which the sample size results in a lower number of pairs of identical sequences, aggregating groups can be a valuable strategy to reduce uncertainty. Within our WA sequencing dataset, we find that assessing spread between two age groups requires around $10^2$–$10^3$ sequences, whereas nine age groups increases the number of sequences required to $10^4$–$10^5$ (Supplementary Fig. 19).

Extending the analysis to pairs of sequences separated by a greater Hamming distance could also increase the statistical power by increasing the amount of data available, in particular for pathogens characterized by a higher mutation rate (Extended Data Fig. 9). However, increasing the threshold comes at the cost of diluting the signal by including pairs that are less epidemiologically linked in the analysis, therefore introducing more noise (Extended Data Fig. 9). Other factors, such as the rate of mixing between groups or the natural history of the infection, can impact the optimal threshold.

## Perspectives

Large-scale pathogen genome sequencing provides an incredible opportunity to understand where and how transmission is occurring. The computational cost of existing methods that rely on inferring the phylogenetic tree has limited their ability to elucidate fine-scale transmission patterns. Analysing datasets that are orders of magnitude larger than those enabled by existing tree-based methods could provide insights into disease spread at finer granularity, such as among more numerous and smaller population groups. Here we show that a simple count-based metric based on pairs of identical pathogen sequences with detailed linked metadata can provide unique insights into the determinants of SARS-CoV-2 transmission. Future work investigating how to better describe asymmetry in transmission between groups and how to infer group-level contributions to epidemic growth from such data are a promising research direction. This shows that relying on pairs of identical or nearly identical pathogen sequences along with fine-grained metadata is valuable to understand how and where transmission is happening. By providing scalable new tools to understand detailed pathogen spread patterns, we believe that this work represents an important development to guide future epidemic control efforts.

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

## Methods

### Data sources and preprocessing

**Sequence data and metadata.** We analysed 116,791 SARS-CoV-2 sequences from Washington state genomic sentinel surveillance system[45] sampled between 1 March 2021 and 31 December 2022. Sequence metadata are collated by the Washington State Department of Health and include sample collection date, symptom onset date, de-identified patient ID, county of home location, postcode of home location, age group (0–9 years, 10–19 years, 20–29 years, 30–39 years, 40–49 years, 50–59 years, 60–69 years, 70–79 years and 80 years and above) and vaccination status upon positive test. For patients with multiple sequences in the database (2,309 out of 114,306 patients), we restricted our analysis to the earliest sequence collected. Among these 114,306 sequences, the age information is missing for 1 sequence, the county information for 659 sequences and the postcode information for 1,011 sequences.

Consensus sequences are extracted from the GISAID EpiCoV database[46,47] and curated using the Nextstrain nCoV ingest pipeline[48]. We discard sequences with undefined Nextstrain clade assignments (8 sequences out of 114,306). This leaves us with 114,298 sequences, with 114,297 sequences gathered from patients with known age, 113,639 sequences gathered from patients with known county of home location and 113,287 sequences gathered from patients with known postcode of home location. In total, 96% of sequences have coverage in greater than 90% of the genome. We match postcodes to zip code tabulation areas (ZCTAs). For postcodes that do not have a ZCTA with the same name, we manually match them by looking at ZCTA boundaries. All analyses at the postcode level use ZCTA metadata information. We extracted the postcodes of WA prison facilities online[49].

**Pairwise genetic distances.** We compute pairwise genetic distances between Washington state sequences with the ape R package[50] using Hamming distances. To avoid unnecessary computational costs, we compare only sequences belonging to the same Nextstrain clade[51] and generate one distance matrix per clade. We do not expect the clade definition to impact the pairs of identical sequences that we identified as pairs of identical sequences should always belong to the same clade.

Generating pairs of identical sequences from a sequence data file is the most computationally expensive step in this analysis. To provide context, generating a distance matrix from 1,000 sequences takes 33 s, while 10,000 sequences takes 1 h 37 min, on 1 core of an Apple M2 chip. Generating the full distance matrix for the analysis set of 113,287 sequences took around 96 hours of compute time readily parallelized across a compute cluster. More efficient software tools can significantly bring that compute time down (for example, 1.14 h with pairsnp[52]).

**Workflow data.** We use data describing the daily number of commuters between each WA county from the American Community Survey (2016–2020)[22]. This dataset provides the number of directed commuting flows between residence and workplace counties. We use the number of commuting flows between counties to compute the RR of commute between two regions (see below).

**Mobile phone location data.** We obtain mobile device location data from SafeGraph (https://safegraph.com/), a data company that aggregates anonymized location data from 40 million devices, or approximately 10% of the US population, to measure foot traffic to over 6 million physical places (points of interest (POIs)) in the United States. Following a previous study[53], we estimate movement within and between counties in Washington from January 2020 to June 2022, using SafeGraph's Weekly Patterns dataset, which provides weekly counts of the total number of unique devices visiting a point of interest (POI) from a particular home census block group (CBG). POIs are fixed locations, such as businesses or attractions. A visit indicates that a device entered the building or spatial perimeter designated as a POI. The home location of a device is defined as its common nighttime (18:00–07:00) CBG for the past 6 consecutive weeks. We restrict our dataset to POIs that have been tracked by SafeGraph since December 2019. To measure movement within and between counties, we extract the home CBG of devices visiting POIs in each week and limit the dataset to devices with home CBGs in the county of a given POI (within-county movement) or with home CBGs in counties outside of a given POI's county (between-county movement). To adjust for variation in SafeGraph's device panel size over time, we divide each county's census population size by the number of devices in SafeGraph's panel with home locations in that county each month and multiply the number of weekly visitors by that value. For each mobility indicator, we sum adjusted weekly visits across POIs from March 2021 to June 2022. We use the number of visits between counties to compute the RR of movement from mobile phone data between two regions (see below).

To explore potential geographical biases in the mobility data, we divided the weekly number of devices residing in each county by the weekly number of devices residing in WA state (observed proportions) and compared these values to expected proportions based on county and state census population sizes during 2020–2022. SafeGraph's panel consistently captured 2–5% of each county's population, with strong correlations between device counts and census population sizes (Spearman's $\rho = 0.99$; Supplementary Fig. 20). We estimated county-level bias as the observed proportion of devices tracked by SafeGraph in individual counties relative to WA state minus the expected proportion based on census population sizes. Annual bias estimates for individual counties ranged from −2.2% to 1.7%, with no clear trend of over- or under-representation by population size or urban–rural classification (Supplementary Fig. 21). Although the most populous counties in WA state tend to have greater absolute bias, large counties are both under-represented and over-represented in the SafeGraph dataset (Supplementary Fig. 21). For example, the most populous county in WA, King County, was slightly under-represented each year (−2.2% to −1.6% bias; green negative outlier in Supplementary Fig. 21), while three of the other top five largest counties (Clark, Pierce and Spokane) were slightly over-represented (1.1% to 1.7% bias; pink, blue and goldenrod positive outliers in Supplementary Fig. 21). Our method for estimating geographical bias is based on SafeGraph's Google Co-Lab Notebook on Quantifying Bias[54].

**Social-contact data.** We use synthetic social contact data for WA generated previously[34] based on reconstructing synthetic populations of interacting individuals using WA population demographics. They describe the per-capita probability for an individual of age $i$ of interacting with someone of age $j$ during a day.

### Quantifying connectivity between groups

**From genetic data.** To quantify connectivity from genetic data, we compute the RR for sequences separated by a given genetic distance of being in given subgroups of the population. Let $n$ denote the number of sequences included in the study and $H_{i,j}$ the Hamming distance between sequences indexed by $i$ and $j$. Let $S_i$ denote the subgroup of sequence $i$. We introduce $n_{A,B}^d$ the number of Hamming distance matrix elements (excluding the diagonal) equal to $d$ where sequence $i$ belongs to group $A$ and sequence $j$ belongs to group $B$.

$$n_{A,B}^d = \sum_{i=1}^{n} \sum_{j=1}^{n} 1_{\{i \neq j\}} \, 1_{\{H_{i,j}=d\}} \, 1_{\{S_i=A\}} \, 1_{\{S_j=B\}}$$

where $X \rightarrow 1_X$ is the indicator function which is equal to 1 if $X$ is true and 0 otherwise.

Let $n_{A,\bullet}^d = \sum_B n_{A,B}^d$ and $n_{\bullet,\bullet}^d = \sum_A \sum_B n_{A,B}^d$.

We derive the RR $RR_{A,B}^d$ for sequences separated by a genetic distance $d$ of being observed in subgroups $A$ and $B$ compared to what is expected from the sequencing effort in the different subgroups of the population as:

$$RR_{A,B}^d = \frac{n_{A,B}^d / n_{A,\bullet}^d}{n_{B,\bullet}^d / n_{\bullet,\bullet}^d} = \frac{n_{A,B}^d \times n_{\bullet,\bullet}^d}{n_{A,\bullet}^d \times n_{B,\bullet}^d} \qquad (1)$$

The numerator $n_{A,B}^d / n_{A,\bullet}^d$ corresponds to the proportion of the pairs where the subgroup of sequence $i$ is $A$ that are occurring with the $B$ group.

The denominator $n_{B,\bullet}^d / n_{\bullet,\bullet}^d$ is a normalization factor quantifying the contribution of group $B$ to the total number of pairs separated by a Hamming distance of $d$. The ratio between these two quantities therefore quantifies the extent to which pairs of sequences observed in groups $A$ and $B$ are enriched compared with the number of sequences observed in these groups.

We used a subsampling strategy to compute CIs around these RRs. Bootstrapping (random sampling with replacement) would result in comparing sequences with themselves and therefore lead to biased upwards RRs of observing identical sequences in the same group. To avoid this, we used a subsampling strategy (random sampling without replacement) with a 80% subsampling rate (1,000 replicate subsamples).

We provide the tools to compute this RR metric from user-provided sequence and metadata files in the GitHub repository associated with this Article[55,56].

**From mobility data.** To quantify connectivity from mobility data, we compute the RR of movement between two geographical locations. Both the mobile phone and commuting mobility data provide directed flows between WA counties. Let $w_{A \to B}$ denote the number of commuters reported in the commuting data (respectively the number of visits for the mobile phone mobility data) whose home residence is in county $A$ and who work in county $B$ (respectively for which a visit in county $B$ is reported). We compute the total movement flow between counties $A$ and $B$ as:

$$w_{A,B} = w_{A \to B} + w_{B \to A}$$

We then calculate the RR $RR_{A,B}^{\text{mobility}}$ of movement between counties $A$ and $B$ as:

$$RR_{A,B}^{\text{mobility}} = \frac{w_{A,B} \times w_{\bullet,\bullet}}{w_{A,\bullet} \times w_{B,\bullet}}$$

where $w_{X,\bullet} = \sum_Y w_{X,Y}$ and $w_{\bullet,\bullet} = \sum_{X,Y} w_{X,Y}$.

We compute a similar statistic by aggregating counties at the regional level (Supplementary Fig. 8).

**From social contact data.** To quantify connectivity from social contact data, we compute the RR of contact between two age groups. Mistry et al.[34] estimated the average daily number of contacts $M_{i,j}$ that individuals of age $i$ have with individuals of age $j$ (considering one-year age bins). As we are interested in the age groups available in the sequence metadata, we reconstruct the average daily number of contacts $c_{A,B}$ that individuals within age group $A$ have with individuals in age group $B$ as:

$$c_{A,B} = \frac{\sum_{i \in A} \sum_{j \in B} M_{i,j} \times n_i}{\sum_{i \in A} n_i}$$

where $n_i$ is the number of individuals of age $i$. We can then derive the total daily number of contacts between age groups $A$ and $B$ as $\Gamma_{A,B} = c_{A,B} \times N_A$ where $N_A$ is the number of individuals in age group $A$. We then compute the RR $RR_{A,B}^{\text{contacts}}$ for a contact of occurring between age groups $A$ and $B$ compared to what we expect if contacts were occurring at random in the population as:

$$RR_{A,B}^{\text{contacts}} = \frac{\Gamma_{A,B} \times \Gamma_{\bullet,\bullet}}{\Gamma_{A,\bullet} \times \Gamma_{B,\bullet}}$$

where $\Gamma_{A,\bullet}$ is the total daily number of contacts involving individuals within age group $A$ and $\Gamma_{\bullet,\bullet}$ is the total daily number of contacts in the population.

**Studying directionality in transmission**
We use the timing of sequence collection to understand directionality in transmission.

**From sequence collection dates.** We introduce $t_x$ as the time at which the sequence $x$ was collected. Let $I_{A,B}$ denote the ensemble of pairs of identical sequences observed in groups $A$ and $B$.

$$I_{A,B} = \{(a, b) \in I_{A,B} | |t_a - t_b| > 0\}$$

therefore denotes the subset of these pairs with different sequence collection dates. We compute the proportion $p_{A \to B}$ of pairs consistent with the transmission direction $A \to B$ as:

$$p_{A \to B} = \frac{\#(\{(a, b) \in I_{A,B} | t_a < t_b\})}{\#(I_{A,B})}$$

where $\#(X)$ is the cardinal of $X$.

We also report 95% binomial CIs around these proportions.

**From symptom onset dates.** The delay between infection and sequence collection can be impacted by healthcare-seeking behaviours and access to testing, which might vary across age groups, geographical locations and time periods. If the distribution of the delay until testing differs between two subgroups $A$ and $B$, the proportion of pairs of identical sequences $p_{A \to B}$ that are first collected in group $A$ will both reflect the timing of infections and healthcare-seeking behaviours. When available, symptom onset dates should be less impacted by healthcare-seeking behaviours.

Among the 113,638 SARS-CoV-2 sequences with associated age group and county of home location information, symptom onset dates are available for 34,167 of them (30%). The availability of symptom onset information is susceptible to be impacted by individual demographic profiles (such as age), which could result in sequences with symptom onset information not being representative of all the sequences available. To avoid this, we impute missing symptom onset dates based on the empirical delay distribution between symptom onset and sequence collection (computed from individuals with known symptom onset dates) stratified by age group, time period and EWA/WWA region (Supplementary Fig. 22). Out of the sequences with known symptom onset dates, the absolute value of the delay between sequence collection and reported symptom onset is strictly greater than 30 days for 192 of them (<0.6%). We discarded these sequences in the computation of the symptom onset to sequence collection delay and considered that they were equivalent to sequences with missing symptom onset information (and therefore imputed their symptom onset dates). We generate 1,000 imputed datasets. For each of these imputed datasets, we compute the proportion $p_{A \to B}^{\text{sympto}}$ of pairs with symptom onset dates occurring first in group $A$ among pairs of identical sequences in groups $A$ and $B$ with distinct symptom onset dates. We then report the median across these 1,000 imputed datasets. We also generate a measure of uncertainty by computing on each of the imputed datasets 95% binomial CIs around the proportion $p_{A \to B}^{\text{sympto}}$. We then report an uncertainty range around these proportion by using the minimum lower bound of the 95% CI and the maximum upper bound of the 95% CI across the imputed datasets.

**Spatial analyses**
**Spatial extent of clusters.** We reconstruct clusters of identical sequences from the pairwise genetic distance matrices[13]. Supplementary Fig. 23 depicts the typical size and duration of these clusters of identical sequences throughout the study period. To assess the spatial and

temporal signal in clusters of identical sequences, we evaluate how the spatial extent of a cluster (summarized by its radius) evolves over time. For each cluster, we define primary sequences as the cluster's earliest collected sequence. We then define the cluster's primary ZCTAs as the ZCTAs of its primary sequences. We exclude clusters with ambiguous primary ZCTA (several primary ZCTAs) from this analysis. We define the radius of a cluster at a given time as the maximum distance between the primary ZCTA and the ZCTAs of the sequences collected by that time. We also compute the time required for sequences to be collected outside the primary ZCTA and primary county (using a similar definition as for the primary ZCTA). We report the mean cluster radius and the proportion of clusters remaining within the same geographical unit (ZCTA and county) as a function of the time since collection of the first sequence within a cluster. We generate 95% CI using a bootstrap approach with 1,000 replicates.

We compare the observed cluster radius and the observed proportion of clusters remaining within the same geographical unit to those expected from a null distribution assuming no spatial dependency between sequences within a cluster of identical sequences. We simulate a null distribution by randomly permuting the geographical locations of the WA sequences and recomputing our statistics of interest (cluster radius, proportion of clusters within the same county and proportion of clusters within the same ZCTA).

**Impact of counties' adjacency status.** We compare the RRs of observing identical sequences between two counties depending on counties' adjacency status (within the same county, between adjacent counties and between non-adjacent counties) using Wilcoxon signed-rank tests.

**Impact of distance between counties.** We examine how the RRs of observing identical sequences in two distinct counties compare with the geographical distance between counties' centroids. We summarize this trend by reporting the LOESS curves with 95% CIs between the log RRs and the distance in kilometres.

**Mapping using MDS.** We evaluate the extent to which patterns of association obtained when looking at the location of pairs of identical sequences are consistent with global spatial structure. To do so, we performed non-metric MDS (NMDS) based on the matrix of RR of observing pairs of identical sequences between two counties. We restrict our analysis to the subset of counties for which there was always at least five pairs of identical sequences observed with the other counties in the subset. This is done to remove potential noise associated with low number of pairs observed. As the NMDS algorithm requires a measure of similarity between counties, we define the similarity $s_{A,B}$ between counties $A$ and $B$ as:

$$s_{A,B} = e^{-RR^0_{A,B}}$$

We perform two-dimensional NMDS using the vegan R package[57].

**Transmission direction: EWA and WWA.** We evaluate whether the timing of identical sequence collection is consistent with transmission rather occurring from WWA to EWA or EWA to WWA. We define four time periods corresponding the four epidemic waves experienced by WA during our study period (Supplementary Fig. 24): wave 4, March 2021–June 2021; wave 5, July 2021–November 2021; wave 6, December 2021–February 2022; and wave 7, March 2022–August 2022. For each of these time periods, we compute the proportion of pairs of identical sequences first collected in EWA among pairs of identical sequences observed in both EWA and WWA that were not collected on the same day. We report 95% binomial proportion CI around these proportions.

To examine whether our conclusions could be explained by differences in testing behaviours between EWA and WWA, we conduct a sensitivity analysis by imputing the date of symptom onset.

## Mobility analyses

**Relationship with mobility data.** We compute the Spearman correlation coefficient between the RR of observing identical sequences between two counties and the RR of movement between two counties (both from mobile phone derived and commuting mobility data) as well as the geographical distance between counties' centroids. We determine the percentage of variance in the genetic data explained by the mobility data by fitting GAMs predicting the RR of observing identical sequences based on the RR of movement between two counties, both on a logarithmic scale, using a thin-plate smoothing spline with 5 knots. For the GAM analyses, we remove pairs of counties for which the number of pairs of identical sequences or the total mobility flow is equal to 0, which ensures that both the RR of observing identical sequences and the RR of movement are strictly positive. We also fit a GAM between the RR of observing identical sequences between two counties (on a logarithmic scale) and the distance between counties centroids. We repeat these analyses at the regional level, instead of at the county level.

**Outliers from that relationship.** We define outliers in the relationship between genetic and mobility data as pairs of counties for which the absolute value of the scaled Pearson residuals of the GAM is greater than 3. As we expect RRs computed from a low number of pairs of identical sequences to be more noisy, we focus on pairs of counties for which at least 100 pairs of identical sequences are observed throughout the study period.

## Spread between male prison postcodes

**Centrality analysis.** We characterize transmission between the ten postcodes with male state prisons by performing a network centrality analysis. We consider a network with ten nodes corresponding to these different postcodes. We define the weight of each edge as the RR of observing identical sequences between the two postcodes that define the nodes connected by this edge. This results in a fully connected network. For each node (postcode with a male state prison), we compute eigenvector centrality scores using the R igraph package. This centrality score measures a node's influence in the network: nodes have higher scores when they are connected to other influential nodes.

**Shared big clusters between postcodes.** We define large clusters of identical sequences in the prison networks as clusters of identical SARS-CoV-2 sequences (1) that are observed in at least two postcodes with male prisons and (2) with at least 15 sequences in prison postcodes.

## Age analyses

**Relationship with social contact data.** We quantify the association between the RR $RR^0_{A,B}$ of observing identical sequences between two age groups $A$ and $B$ and the RR of contacts $RR^{contacts}_{A,B}$ between these two groups using a GAM on a logarithmic scale. We report the percentage of variance in the RR of identical sequences explained by the RR of contact from the GAM. We also report the Spearman correlation coefficient between $RR^0_{A,B}$ and $RR^{contacts}_{A,B}$.

**Age-specific transmission across scales.** To understand how age-specific transmission patterns vary across spatial scales, we compare the RR of observing identical sequences between age groups using all pairs of identical sequences, using only pairs of identical sequences in different postcodes and using only pairs of identical sequences in different counties.

**Typical direction of spread between ages.** We use sequence collection dates to explore transmission direction between age groups across four periods (March 2021–June 2021, July 2021–November 2021, December 2021–February 2022 and March 2022–August 2022).

To facilitate the interpretation of these results, we introduce an earliness score that measures the contribution of a given age group to transmission to other age groups. For an age group $A$, this score is equal to the proportion of pairs of identical sequences first observed in age group $A$ among all pairs of identical sequences observed in age groups $A$. We also report the 95% binomial CI around this score.

To explore whether our conclusions could be explained by differences in testing behaviours between age group, we conduct a sensitivity analysis by imputing the dates of symptom onset and using symptom onset dates instead of sequence collection ones (Supplementary Fig. 15). We also compute earliness scores on each of the 1,000 datasets with imputed symptom onset dates using the same definition as that based on sequence collection dates. We then report the median earliness score across all 1,000 datasets as well as an uncertainty range defined as the minimum lower bound and the maximum upper bound of the 95% binomial CI around this score for each of the imputed dataset.

**RR between vaccination groups.** Available matched patient information include details regarding the individuals' vaccination statuses upon positive test: No valid vaccination record (denoted unvaccinated); completed primary series (denoted vaccinated); and completed primary series with an additional dose (denoted boosted).

Here, we use this information to quantify mixing between groups characterized by their vaccination status. We focus on the mixing between vaccination groups within age groups to avoid biases coming from age group and vaccination status being correlated. Among sequences collected within each period (4 waves) and age groups in decade, we compute the RR of observing identical sequences between vaccination groups. We included the boosted vaccination group for wave 6 (Omicron BA.1 wave) only for age groups older than 10 years, and included the boosted vaccination group for wave 7 for the 0–9 year age group. We included the 0–9 year age group in our analysis only from wave 6 (Omicron BA.1 wave) and the 10–19 year age group from wave 5 (Delta wave).

To quantify the tendency of individuals of transmitting to individuals with the same vaccination status, we compute for each vaccination groups $(V_1, V_2)$ the ratio $RR_{V_1,V_2}/RR_{V_1,V_1}$. Values lower than 1 indicate that the enrichment of pairs of identical sequences is greater within the same vaccination group than between different vaccination groups. Such values suggest assortativity in mixing patterns between vaccination groups.

## Sensitivity analysis: direction analysis

In the former paragraphs, we describe an approach based on the timing of pairs of identical sequences to better understand the typical transmission direction between groups. The interpretation of this pair-based analysis is complicated by several factors. First, clusters of identical sequences can span more than two groups. Second, even in instances where clusters only span two groups, counting pairs could improperly capture transmission direction, for example if local transmission of the cluster is occurring within the two groups. We implemented this pair-based approach as an intuitive exploration of whether the timing of sampling of identical sequences might provide some signal about transmission direction. This pair-based approach is crude but yet interesting as we do expect groups that tend to be the source to more often be observed first within clusters or pairs of identical sequences. As a sanity check, we perform a sensitivity analysis relying on clusters of identical sequences that are observed only within two groups. We define the source group as the group of the earliest collected sequence within the cluster of identical sequences. We remove ambiguous clusters, meaning clusters with two potential source groups, from the analysis. For clusters observed in groups $A$ and $B$, we compute the proportion of clusters with source group $A$. We refer to this proportion as 'proportion from clusters' to distinguish it from the 'proportion from pairs' that we use in our main analysis. We compute the 95% CI around the proportion from clusters. This proportion from clusters should be more robust that the proportion from pairs but tends to be noisier as we are computing the proportion from less observations. We then compare the proportion obtained from pairs and from clusters. We compute the Spearman correlation coefficient between these two proportions using all pairs of groups or only pairs of groups for which the CIs do not contain 50% for the two proportions.

## Link between mutations and generations

In this section, we derive the probability distribution of the number of mutations $M_{AB}$ separating the consensus genomes of two infected individuals $A$ and $B$ conditional on the number of transmission generations $G_{AB}$ separating them.

**Generation time distribution.** We assume that the generation time (that is, the average duration between the infection time of an infector and an infectee) follows a Gamma distribution of shape $\alpha$ and scale $\beta$. The time between $g$ generations then follows a Gamma distribution of shape $g \times \alpha$ and scale $\beta$ assuming independence of successive transmission events. Let $f_{\alpha,\beta}(\cdot)$ denote the probability density function of a Gamma distribution of shape $\alpha$ and scale $\beta$.

**Mutations events.** Let $M_{AB}$ denote the number of mutations separating their infecting viruses. Let $\mu$ denote the mutation rate of the virus (in mutations per day). Let $T_{AB}^{\mathrm{evo}}$ denote the evolutionary time separating $A$ and $B$ (in days).

Assuming a Poisson process for the occurrence of mutations, we have:

$$M_{AB} \sim \mathcal{P}(\mu \times T_{AB}^{\mathrm{evo}})$$

**Probability distribution.** Let $G_{AB}$ denote the number of generations separating two infected individuals $A$ and $B$ belonging to the same transmission chain.

$$\begin{aligned}
P[M_{AB}=m|G_{AB}=g] &= \int_{t_{AB}^{\mathrm{evo}}=0}^{+\infty} P[M_{AB}=m|G_{AB}=g, T_{AB}^{\mathrm{evo}}=t_{AB}^{\mathrm{evo}}] \\
&\quad \times p(t_{AB}^{\mathrm{evo}}|G_{AB}=g)\, dt_{AB}^{\mathrm{evo}} \\
&= \int_{t_{AB}^{\mathrm{evo}}=0}^{+\infty} P[M_{AB}=m|T_{AB}^{\mathrm{evo}}=t_{AB}^{\mathrm{evo}}] \times f_{\alpha g,\beta}(t_{AB}^{\mathrm{evo}})\, dt_{AB}^{\mathrm{evo}} \\
&= \int_{t_{AB}^{\mathrm{evo}}=0}^{+\infty} \frac{(\mu t_{AB}^{\mathrm{evo}})^m \times e^{-\mu t_{AB}^{\mathrm{evo}}}}{m!} \times \frac{\beta^{\alpha g} \times (t_{AB}^{\mathrm{evo}})^{\alpha g-1} \times e^{-\beta t_{AB}^{\mathrm{evo}}}}{\Gamma(\alpha g)}\, dt_{AB}^{\mathrm{evo}} \\
&= \frac{\mu^m\, \beta^{\alpha g}}{m!\ \Gamma(\alpha g)} \int_{t_{AB}^{\mathrm{evo}}=0}^{+\infty} (t_{AB}^{\mathrm{evo}})^{m+\alpha g-1} \times e^{-(\mu+\beta)\times t_{AB}^{\mathrm{evo}}}\, dt_{AB}^{\mathrm{evo}} \\
&= \frac{\mu^m\, \beta^{\alpha g}}{m!\ \Gamma(\alpha g)} \frac{\Gamma(m+\alpha g)}{(\mu+\beta)^{m+\alpha g}} \int_{t_{AB}^{\mathrm{evo}}=0}^{+\infty} f_{m+\alpha g,\mu+\beta}(t_{AB}^{\mathrm{evo}})\, dt_{AB}^{\mathrm{evo}} \\
&= \frac{\Gamma(m+\alpha g)}{m!\ \Gamma(\alpha g)} \times \left(\frac{\beta}{\beta+\mu}\right)^{\alpha g} \times \left(\frac{\mu}{\beta+\mu}\right)^m
\end{aligned}$$

which is the probability mass function of a negative binomial distribution of parameters:

$$r = \alpha g$$
$$p = \frac{\beta}{\beta+\mu}$$

**Application to SARS-CoV-2.** We compute these probabilities for SARS-CoV-2 considering a mutation rate $\mu = 8.98 \times 10^{-2}$ substitutions per day (32.76 substitutions per year)[58]. We assume that the generation time is Gamma distributed with a mean 5.9 days and s.d. of 4.8 days (ref. 59).

## Performance of the RR framework

We conduct a simulation study to evaluate how our RR framework performs under different sequencing scenarios. We also compare the results obtained from a phylogeographical analysis.

**Simulating synthetic sequence data.** We use ReMASTER[60] to simulate an SEIR epidemic in a structured population with 5 demes, each populated with 100,000 inhabitants. We simulate an epidemic characterized by a basic reproduction number $R_0$ of 2 with a daily time-step. We initiate the simulations by introducing a single infected individual (compartment $I$) in the population (group of index 0). We consider a pathogen with a 3,000 kb genome evolving following a Jukes–Cantor evolution model with a substitution rate of $3 \times 10^{-5}$ substitutions per site per day. After infection, infected individuals enter an exposed (E) compartment during which they are not infectious yet and that they exit at a rate of 0.33 per day. They then enter an infectious (I) compartment where they are infectious that they exit at a rate of 0.33 per day. Sequencing occurs after exit of the I compartment. Given that our RR does not account for directionality in transmission, we considered a scenario with symmetric migration rates. We draw migration rates between demes from a log-uniform distribution of parameters $(10^{-3}, 10^{-1})$.

We then explore two sequencing scenarios. In an unbiased scenario, we assume that each individual has the same probability of being sequenced in each deme. In a biased scenario, we assume that the sequencing probability varies between demes. We draw deme-specific relative sequencing probabilities from a log-uniform distribution of parameters $(10^{-3}, 10^{-1})$. In the unbiased scenario, we fix sequencing probability to the mean of the sequencing probabilities across demes in the biased scenario. We explore different sequencing intensities by scaling these probabilities by different multiplicative factor (Supplementary Table 1): a scaling factor of 0.1 resulting in a mean sequencing probability of 0.43% and a dataset of around 1,700 sequences (used for the DTA analyses); a scaling factor of 0.5 resulting in a mean sequencing probability of 2.16% and a dataset of around 8,600 sequences (used for the RR and the DTA analyses); and a scaling factor of 2 resulting in a mean sequencing probability of 8.66% and a dataset of around 34,500 sequences (used for the RR analyses).

**DTA.** We conduct phylogeographical inference using symmetric DTA[14] using the Bayesian stochastic search variable selection (BSSVS) model implemented in BEAST (v.1.10.4)[61] applied to the synthetic data simulated in our two sequencing scenarios. To isolate the accuracy and precision of the phylogeographical reconstruction, we run our DTA using an empirical tree that is generated directly from ReMASTER simulations. Directly inputting such a tree is not possible in real-world scenarios in which the genealogical tree must be (noisily) estimated from empirical sequence data. In this case, it serves a demonstration of the power of DTA when provided perfect genealogical signal. The empirical tree approach also requires substantially less computation and therefore enabled us to analyse datasets of thousands of sequences using DTA in acceptable time.

Two independent Markov chain Monte Carlo (MCMC) procedures are run for $2.5 \times 10^8$ iterations and sampled every 1,000 iterations. The resulting posterior distributions are combined after discarding initial 10% of sampled trees as burn-in from each of them. We used Tracer (v.1.7)[62] to assess convergence and to estimate the effective sampling size (ESS), ensuring ESS values greater than 200 for each migration rate estimate. We adjust the estimated migration rates by the estimated rate scalar to calculate the per-day rates of transition between demes.

To evaluate how estimating the genealogical tree from empirical sequence data impacts both results and computing times, we perform an additional phylogeographical analysis based on simulated but this time jointly inferring the genealogical tree and migration rates. We run this model for 24 days (corresponding to 475,733,000 MCMC steps), until each migration rate parameter has an ESS greater than 200.

**RR analysis.** We compute the RR of observing identical sequences between two demes $i$ and $j$ and compare these RR to the daily probability $p_{i,j}$ of migration between these two demes which is computed as:

$$p_{i,j} = 1 - \exp(-m_{i,j}) \quad j \neq i$$

$$p_{i,i} = 1 - \sum_{k \neq i} p_{i,k}$$

where $m_{i,j}$ is the migration rate between demes $i$ and $j$. We generate 95% subsampling CIs around the RRs using an 80% subsampling rate.

## Expected relationship with RR of contact

We perform a simulation study to characterize the expected relationship between the RR of observing identical sequences between groups and the RR of contacts between these groups. We illustrate this by looking at transmission between age groups but we expect a similar relationship between the RR of observing identical sequences between regions and the RR of movement between these regions.

To do so, we generate clusters of identical sequences including the age group of the corresponding infected individuals assuming a probability that an infector and an infectee have the same consensus sequences $p$ of 0.7 (close to the value we previously estimated for SARS-CoV-2[13]), a reproduction number $R$ of 1.2 and a sequencing fraction $p_{seq}$ of 0.1.

We use a contact matrix estimated previously[34] to characterize disease transmission between age groups. We assume that the probability $p_{A,B}$ for a contact from an infected individual in age group $A$ to occur with a susceptible individual in age group $B$ is equal to:

$$p_{A,B} = \frac{c_{A,B}}{\sum_{B'} c_{A,B'}}$$

where $c_{A,B}$ is the mean daily number of contacts that an individual in age group $A$ has with individuals in age group $B$. We introduce an age-specific reproduction number ($R_A$) describing the average number of secondary cases infected by a single primary case within age group $A$. As different age groups have different average daily total number of contacts, the age-specific reproduction number varies between age groups. It can be derived as:

$$R_A = \frac{\sum_{B'} c_{A,B'}}{\rho(C)} \times R$$

where $\rho(C)$ is the maximum eigenvalue of the matrix $C = (c_{A,B})$[63]. We then simulate individual clusters of identical sequences using the following steps. First, we initialize clusters by drawing the age of the primary case $A_{primary}$ from a uniform distribution. Second, we simulate clusters as successive infections with identical genomes. At each generation, for each individual infected at the previous generation, let $A$ denote the age of this infectious individual. We use the following procedure:
1. Draw the number of infections with the same genotype: $I \sim \mathcal{P}((p \times R_A)^{-1})$ (Poisson distribution).
2. Draw the age of these individuals: $(A_1, \ldots, A_I) \sim \mathcal{M}(I, n_{age}, (p_{A,1}, \ldots, p_{A,n_{age}}))$ where $\mathcal{M}(n, k, (p_1, \ldots, p_k))$ is a multinomial distribution with $n$ trials, $k$ possible events with probabilities $(p_1, \ldots, p_k)$.
3. Draw the sequencing status of each new cluster member from a Bernoulli distribution of parameter $p_{seq}$.

We end simulations after ten generations to minimize computational costs.

## Dataset size required to compute RRs

We implement a downsampling strategy to understand the amount of sequencing data required to compute RR estimates. We consider genome datasets of the following sizes: $\{1 \times 10^2, 2 \times 10^2, 3 \times 10^2, 4 \times 10^2, 5 \times 10^2, 6 \times 10^2, 7 \times 10^2, 8 \times 10^2, 9 \times 10^2, 1 \times 10^3, 2 \times 10^3, 3 \times 10^3, 4 \times 10^3, 5 \times 10^3, 6 \times 10^3, 7 \times 10^3, 8 \times 10^3, 9 \times 10^3, 1 \times 10^4, 2 \times 10^4, 3 \times 10^4, 4 \times 10^4, 5 \times 10^4, 6 \times 10^4, 7 \times 10^4, 8 \times 10^4, 9 \times 10^4, 1 \times 10^5\}$. For each of

these dataset sizes, we generated 100 downsampled datasets from our WA sequencing data. For each of these downsampled datasets, we compute the RR of observing identical sequences between age groups (Supplementary Fig. 25). To understand how the number of groups studied impacts the amount of data required, we also compute RR of observing identical sequences between aggregated age groups:

- 0–39 years and over 40 years for 2 age groups;
- 0–29 years, 30–59 years and over 60 years for 3 age groups;
- 0–19 years, 20–39 years, 40–59 years and over 60 years for 4 age groups;
- 0–9 years, 10–19 years, 20–29 years, 30–39 years, 40–49 years, 50–59 years, 60–69 years, 70–79 years and over 80 (standard definition used throughout the paper) for 9 age groups.

We compute the error between the RR obtained from a subsampled dataset $RR^d$ and the RR from the full dataset $RR^f$ as:

$$\epsilon = \frac{RR^d - RR^f}{RR^f}$$

For each pair of age groups, we then compute the number of pairs of identical sequences required for the error to be below 10%.

### Extending the Hamming distance threshold

In this work, we assess how pairs of infected individuals whose infected genome is separated by 0 mutations can help to characterize population-level transmission patterns. We apply this method to SARS-CoV-2 sequences from WA but our approach should be broadly applicable for epidemics caused by pathogens where the timescale of mutation events is similar to that of transmission events. In this section, we describe a simulation approach to understand how a pathogen's mutation rate could impact the optimal Hamming distance threshold to apply our RR framework.

We implement the same simulation framework as the one described in the section 'Performance of the RR framework'. We consider that each infection has the same probability of being sequenced (equal to 4.33%, corresponding to a sequencing probability scaling factor of 1). We explore a range of scenarios for the pathogen's mutation rate. To do so, we introduce a multiplicative scaling factor for the baseline pathogen's mutation rate ($3 \times 10^{-5}$ substitutions per site per day) with values ranging between 0.1 and 10. For each multiplicative scaling factor, we perform 100 replicate simulations.

For each simulated epidemic, we count the number of pairs separated by less than $d$ mutations between two regions (for $d$ varying between 0 and 10). In certain scenarios (for example, those characterized by a high mutation rate and a low Hamming distance threshold $d$), we sometimes do not observe any pairs less than $d$ mutations away in a specific group. To be able to compute the RR even in those scenarios, we report a modified version of the RR of observing sequences separated by less than $d$ mutations between two regions:

$$\widetilde{RR}^d_{A,B} = \frac{(n^d_{A,B} + 1) \times (n^d_{\bullet,\bullet} + 1)}{(n^d_{A,\bullet} + 1) \times (n^d_{B,\bullet} + 1)}$$

with the same notations as the ones used in the definition of the RR of observing sequences less than $d$ mutations away in groups $A$ and $B$.

We then compute the Spearman correlation coefficient between the RR for pairs of sequences less than $d$ mutations apart of being in two regions and the daily migration probability between these regions. In simulations in which the s.d. of the RR is equal to 0 (all modified RRs have the same value), we assume that the Spearman correlation coefficient is equal to 0 (RRs are not informative about migration rates).

### Ethics approval

The Washington State and University of Washington Institutional Review Boards determined this project to be surveillance activity and exempt from review; the need for informed consent was waived through this determination. Under Washington State IRB Exempt Determination 2020-102, symptom onset date, age group, residence county, residence postcode and vaccination history was provided by the Washington Department of Health from the Washington Disease Reporting System for individuals with linked sequenced SARS-CoV-2 samples from 1 March 2021 to 31 December 2022. Sequencing and analysis of samples from the Seattle Flu Study was approved by the Institutional Review Board (IRB) at the University of Washington (protocol STUDY00006181). Sequencing of remnant clinical specimens at UW Virology Lab was approved by the University of Washington Institutional Review Board (protocol STUDY00000408).

### Reporting summary

Further information on research design is available in the Nature Portfolio Reporting Summary linked to this article.

### Data availability

All 114,552 SARS-CoV-2 genome sequences referenced in this Article are shared in the GISAID EpiCoV database and are available with standard metadata (generally consisting of date of sample collection, state of sample collection and sometimes county of sample collection)[46,47]. GISAID accessions and a sequence-level acknowledgements table are provided in the GitHub repository associated with this manuscript[55,56]. In total, 102,528 of these sequences are available publicly in NCBI GenBank and GenBank accessions are referenced in the acknowledgment table. More detailed metadata curated by Washington State Department of Health (WA DOH) of county, postcode, age group and vaccination status were shared with the Fred Hutchinson Cancer Center under a Data Sharing Agreement for Confidential Data with an associated IRB exempt determination as determined by the Washington State Institutional Review Board. As WA DOH remains the owner of these data, the authors cannot share any portion of this granular metadata directly. Instead, requests to access these data must go directly to WA DOH as described below. Access to more granular metadata is managed by the Washington State Department of Health (WA DOH). Parties interested in reusing these data for their own analyses will need their own data sharing agreements and human subjects review. To initiate a collaboration, interested parties can contact A.B. (alli.black@doh.wa.gov). RR computations in the Article derive entirely from counts of pairs of identical sequences shared between groups. To reproduce analyses, we share counts of pairs of identical sequences between population groups (counties and age groups) obtained from processing these sequence data alongside granular metadata at GitHub (https://github.com/blab/phylo-kernel-public)[55,56]. We also provide a detailed explanation on how to reproduce the analyses using only the data publicly available on NCBI GenBank.

### Code availability

Code to reproduce our analyses is available at GitHub (https://github.com/blab/phylo-kernel-public)[55] and Zenodo (https://doi.org/10.5281/zenodo.14829446)[56]. A step-by-step tutorial describing how to implement our RR approach is available at GitHub (https://github.com/CecileTK/tutorial-rr-identical-sequences)[64].

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

**Acknowledgements** We thank S. Gurrey, L. B. Strick and M. A. Santoro for discussions on the control of SARS-CoV-2 in WA Correction Centers; and all data contributors, including the authors and their originating laboratories responsible for obtaining the specimens, and their submitting laboratories for generating the genetic sequence and metadata and sharing through the GISAID Initiative, on which this research is based. This work is supported by NIH NIGMS (R35 GM119774 to T.B.) and the CDC (CDC-RFA-CK22-2204, Pathogens Genomics Centers of Excellence, contract NU50CK000630). Sequencing of specimens by the Brotman Baty Institute of Precision Medicine was funded by Gates Ventures (Seattle Flu Study award), Howard Hughes Medical Institute (HHMI COVID-19 Collaboration Initiative award) and the CDC (contract number 200-2021-10982). Sequencing of specimens by UW Virology was funded by Fast Grants (award 2244), the CDC (contracts 75D30121C10540 and 75D30122C13720) and WA DOH (contract HED26002). Some of the analyses were completed using Fred Hutch Scientific Computing resources (NIH grants S10-OD-020069 and S10-OD-028685). T.B. is a Howard Hughes Medical Institute Investigator. The findings and conclusions in this report are those of the authors and do not necessarily represent the official position of the US National Institutes of Health or Department of Health and Human Services.

**Author contributions** C.T.-K. and T.B. conceived the study. C.T.-K. developed the methods. C.T.-K., A.C.P. and M.I.P. performed the analyses with input from C.V. and T.B. L.A.F., H.X., K.K., A.W., A.L.G., P.R., J.M.P., A.D., H.H., D.M., P.D., L.G., C.D.F., E.R., J.S., L.S., D.R., A.T., C.Y., F.A. and A.B. collected and curated the data. C.T.-K., A.C.P., M.I.P. and T.B. wrote the manuscript. A.L.G., P.R., L.A.F., A.B. and C.V. edited the manuscript. All of the authors reviewed the final version.

**Competing interests** A.L.G. reports contract testing from Abbott, Cepheid, Novavax, Pfizer, Janssen and Hologic, research support from Gilead, and salary and stock grants for LabCorp, an immediate family member, outside of the described work. The other authors declare no competing interests.

**Additional information**
**Correspondence and requests for materials** should be addressed to Cécile Tran-Kiem.

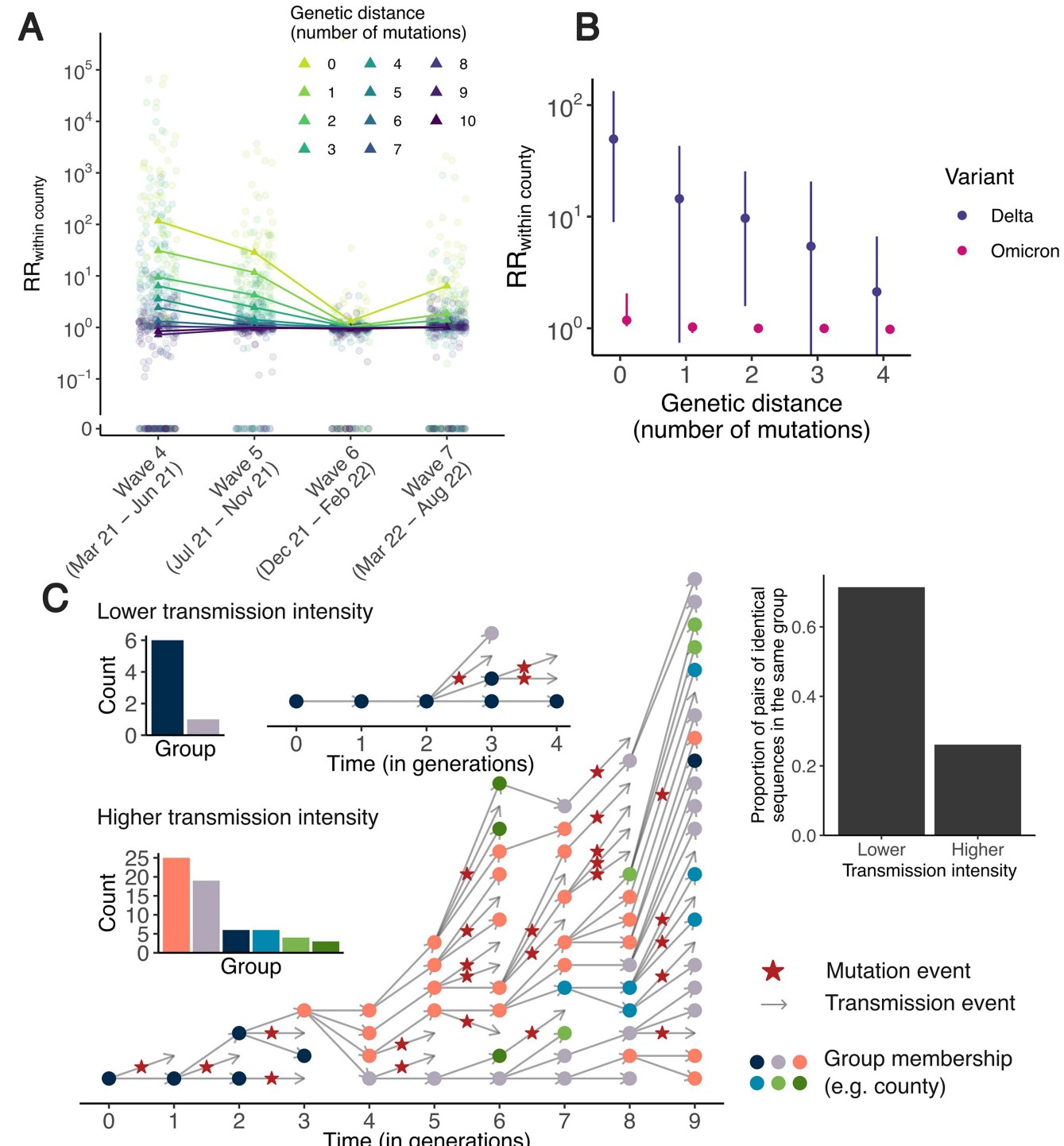

**Extended Data Fig. 1 | The magnitude of the relative risk of observing sequences at a given genetic distance within the same county is impacted by transmission intensity. A**. Relative risk of observing sequences at a given genetic distance within the same county across multiple epidemic waves. We defined waves as: March 2021-June 2021 (Wave 4), July 2021-November 2021 (Wave 5), December 2021-February 2022 (Wave 6) and March 2022-August 2022 (Wave 7). In A, circular points correspond to individuals counties and triangles correspond to the median across counties. **B**. Median relative risk of observing pairs sequences within the same county (with IQR) as a function of genetic distance stratified by variant during Wave 6. **C**. A higher transmission intensity results in larger clusters of identical sequences that tend to be more mixed across groups. In C, the two clusters are simulated using a branching process with mutation[13] by assuming the probability for an infector and an infectee to have the same consensus sequence equal to 0.69 and a probability for an infectee of being in the same groups as its infector of 0.7. We consider a reproduction number of 1.2 for the lower transmission intensity scenario and of 2.0 in the higher transmission intensity scenario.

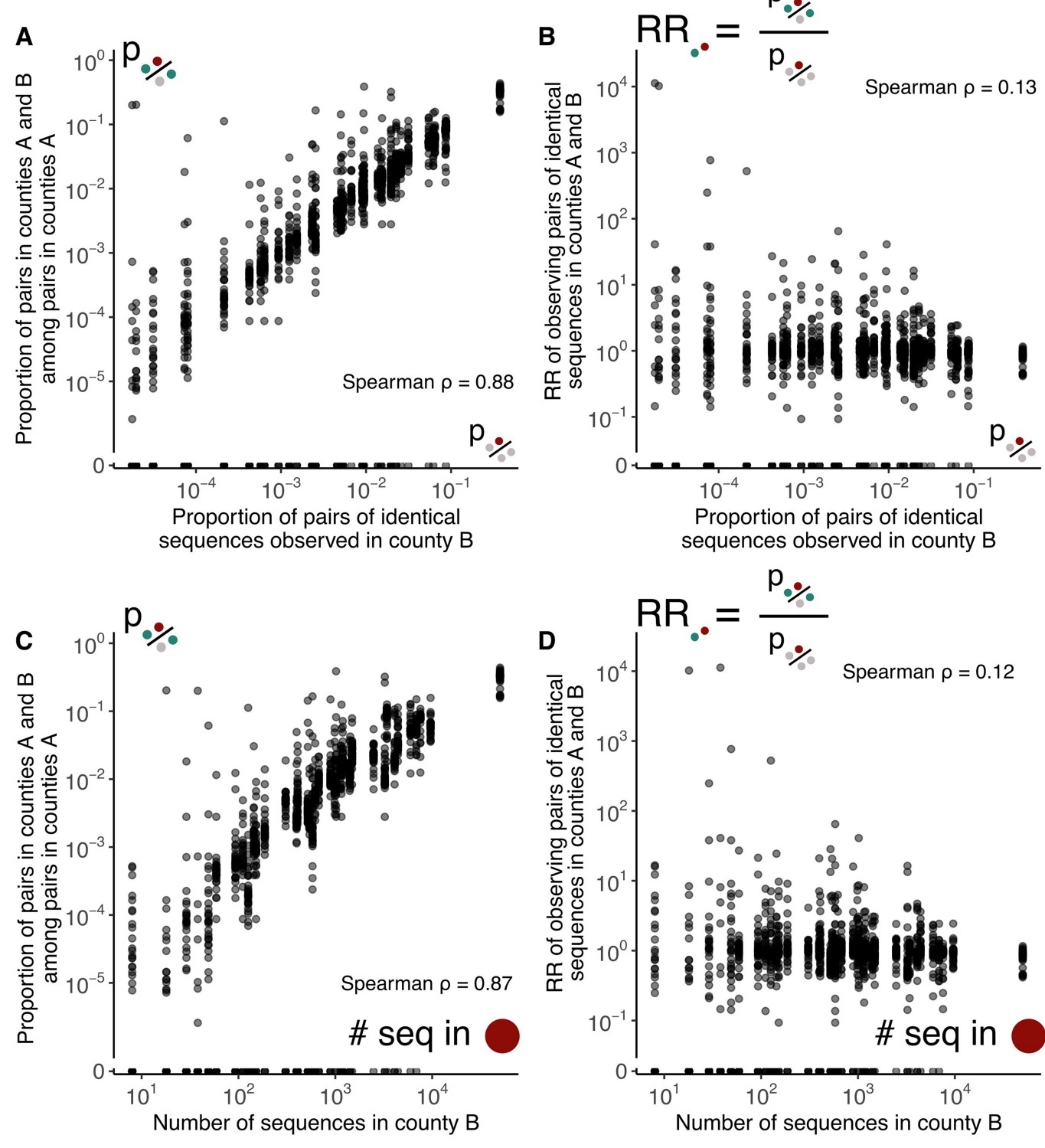

**Extended Data Fig. 2 | Our measure of relative risk corrects for uneven sequencing between regions. A**. Proportion of pairs of identical sequences shared between counties A and B among pairs observed in county A as a function of the proportion of pairs of identical sequences observed in county B. **B**. Relative risk for pairs of identical sequences of being observed in counties A and B as a function of the proportion of pairs of identical sequences observed in county B. **C**. Proportion of pairs of identical sequences shared between counties A and B as a function of the number of sequences available in county B. **D**. Relative risk for pairs of identical sequences of being observed in counties A and B as a function of the number of sequences available in county B.

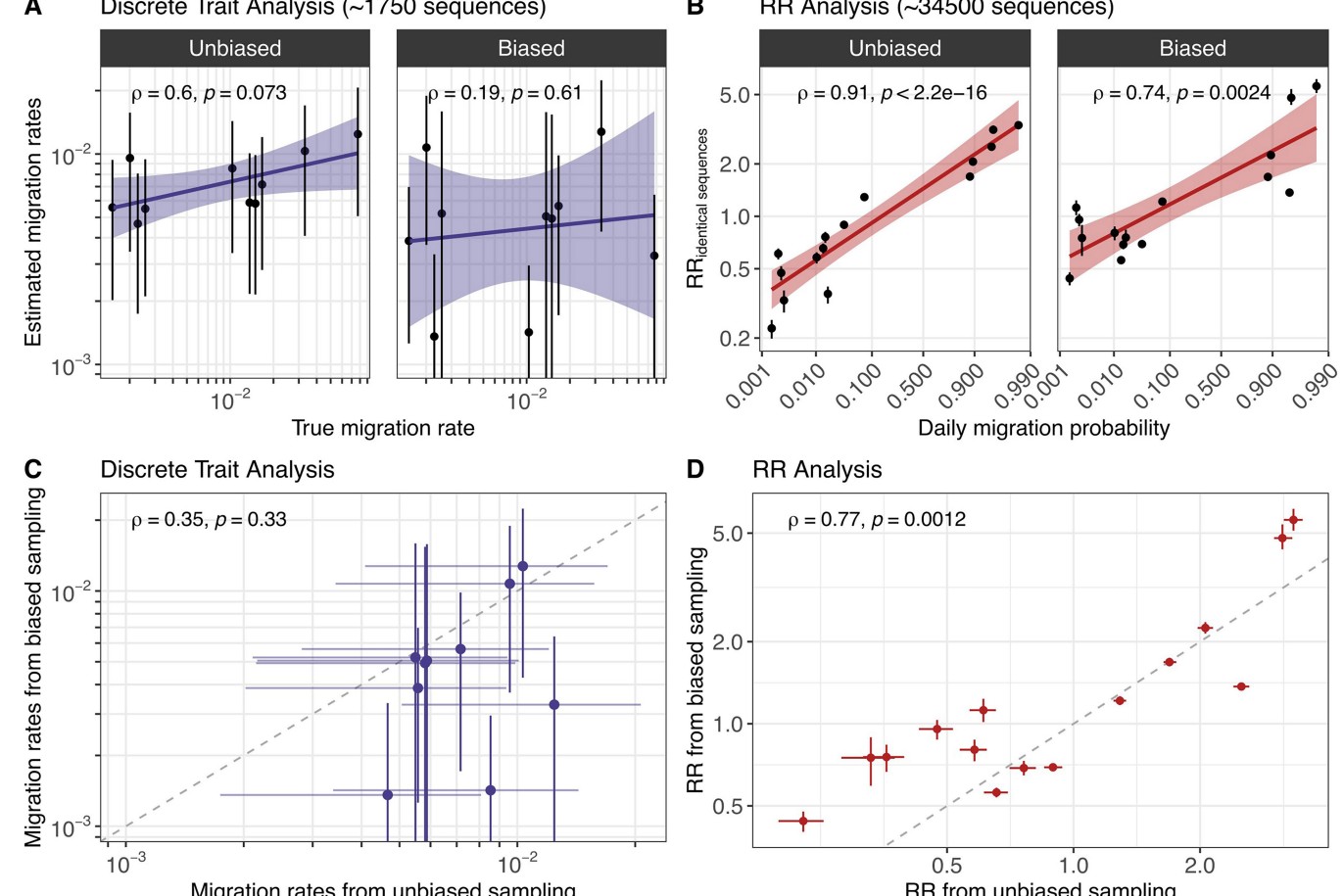

**Extended Data Fig. 3 | Simulation study exploring the impact of sequencing bias on results from a discrete trait analysis and from our RR framework.** **A**. Comparison between migration rates estimated from a discrete trait analysis and the true migration rates used to simulate the sequence data. **B**. Comparison between the relative risk of observing identical sequences between two demes and the weekly migration probability between demes. **C**. Comparison between migration rates inferred from a sequence dataset generated in a biased sampling and an unbiased sampling scenario. **D**. Comparison between the relative risk of observing identical sequences in two groups from a sequence dataset generated in a biased sampling and an unbiased sampling scenario. For the RR, segments indicate 95% subsampling confidence intervals. For the migration rates, segments indicate 95% highest posterior density intervals. For each plot, we indicate the Spearman correlation coefficient (and the associated p-value).

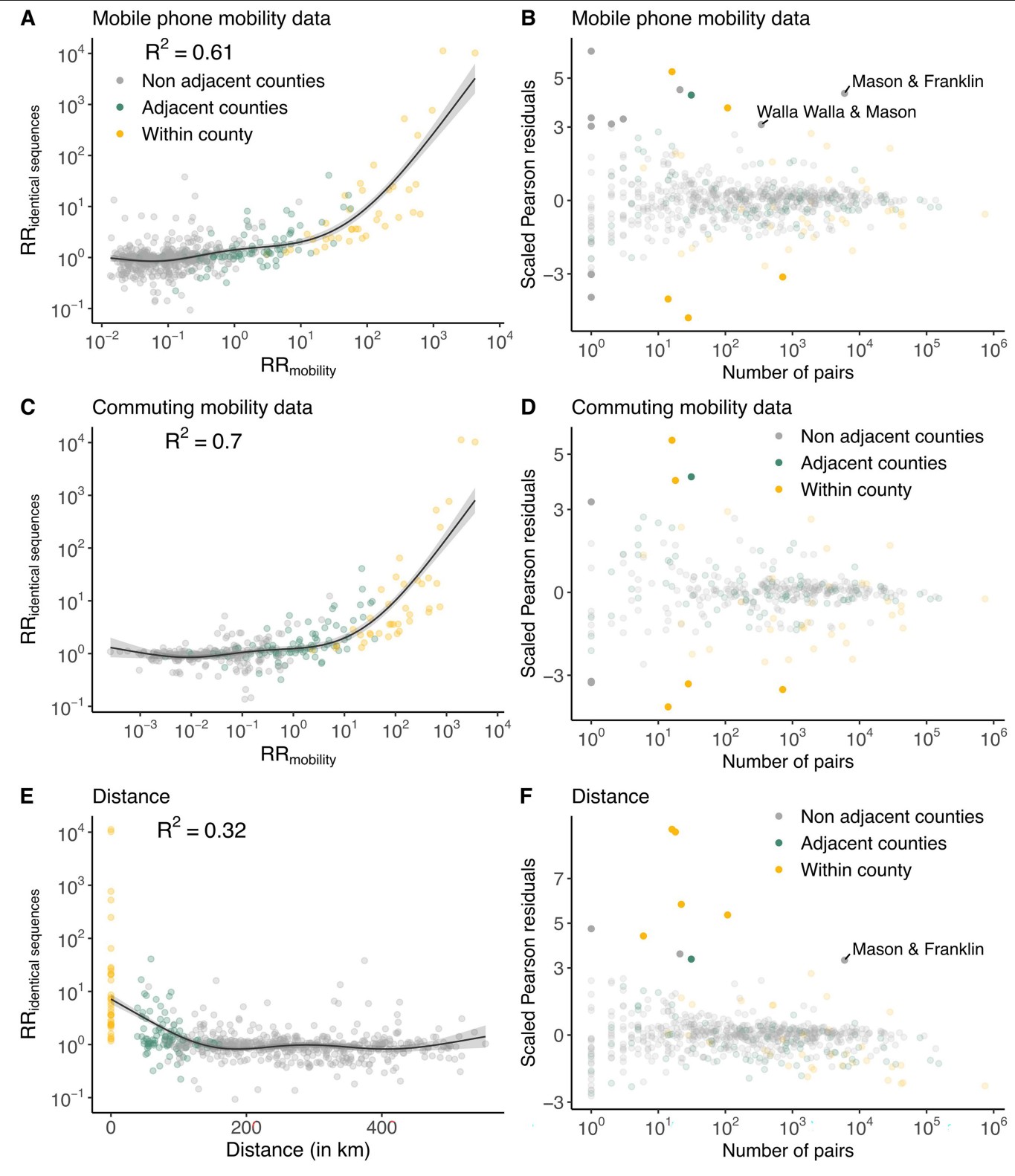

**Extended Data Fig. 4 | Comparison between the relative risk of observing identical sequences and the relative risk of movement at the county level.**
**A**. Relationship between the relative risk of observing identical sequences in two counties and the relative risk of movement between these counties as obtained from mobile phone mobility data. **B**. Scaled Pearson residuals of the GAM plotted in A as a function of the number of pairs of identical sequences observed in pairs of counties. **C**. Relationship between the relative risk of observing identical sequences in two counties and the relative risk of movement between these counties as obtained from workflow mobility data. **D**. Scaled Pearson residuals of the GAM plotted in C as a function of the number of pairs

of identical sequences observed in pairs of counties. **E**. Relationship between the relative risk of observing identical sequences in two counties and the Euclidean distance between counties centroids. **F**. Scaled Pearson residuals of the GAM plotted in E as a function of the number of pairs of identical sequences observed in pairs of counties. In B, D and F, we label pairs of non-adjacent counties sharing at least 100 pairs of identical sequences and for which the absolute value of the Scaled Pearson residual is greater than 3. The trend lines correspond to predicted relative risk of observing identical sequences in two regions from each GAM. $R^2$ indicate the variance explained by each GAM.

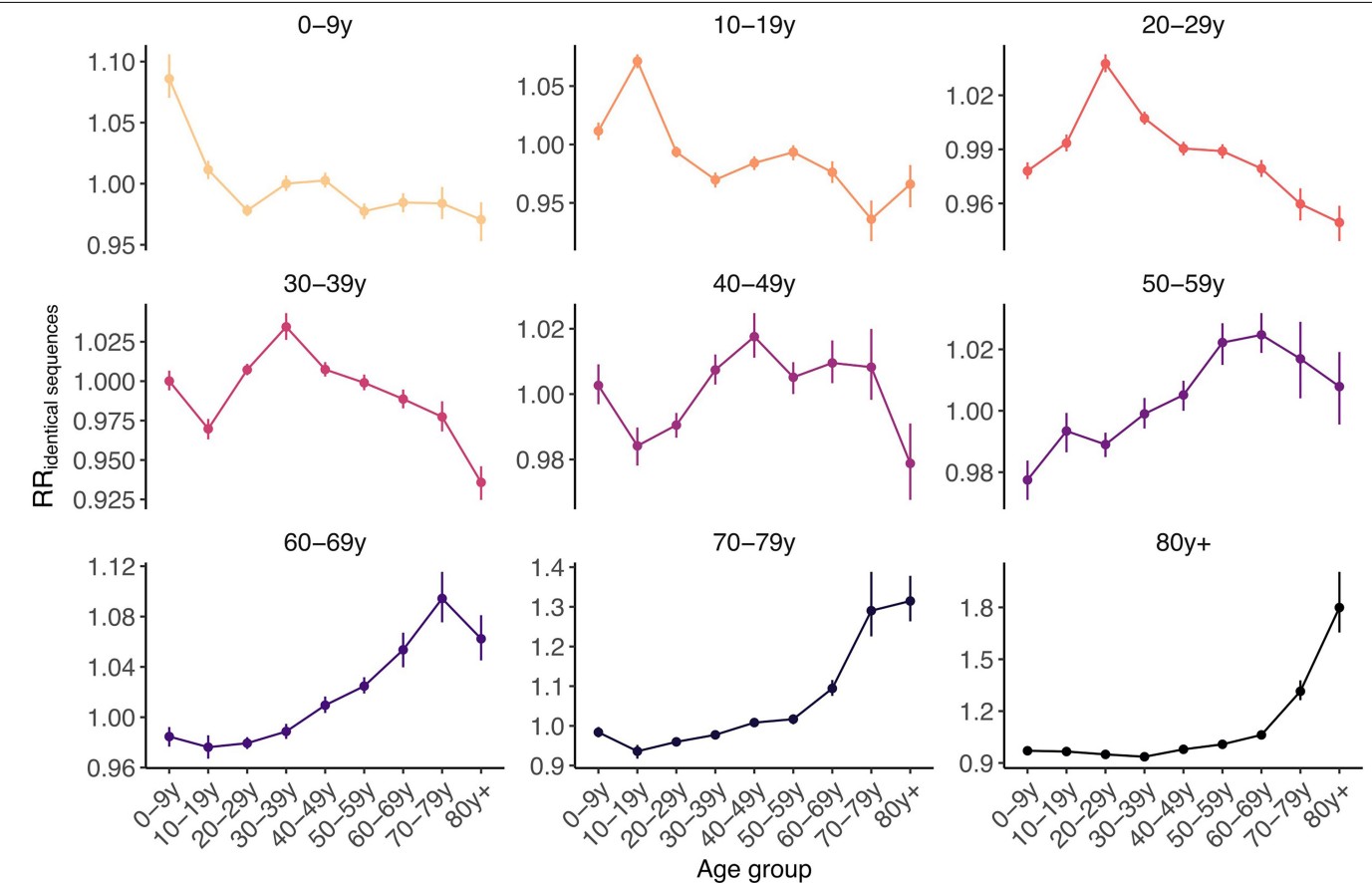

**Extended Data Fig. 5 | Relative risk for pairs of identical sequences of being observed between two age groups.** Vertical segments correspond to 95% confidence intervals obtained through subsampling.

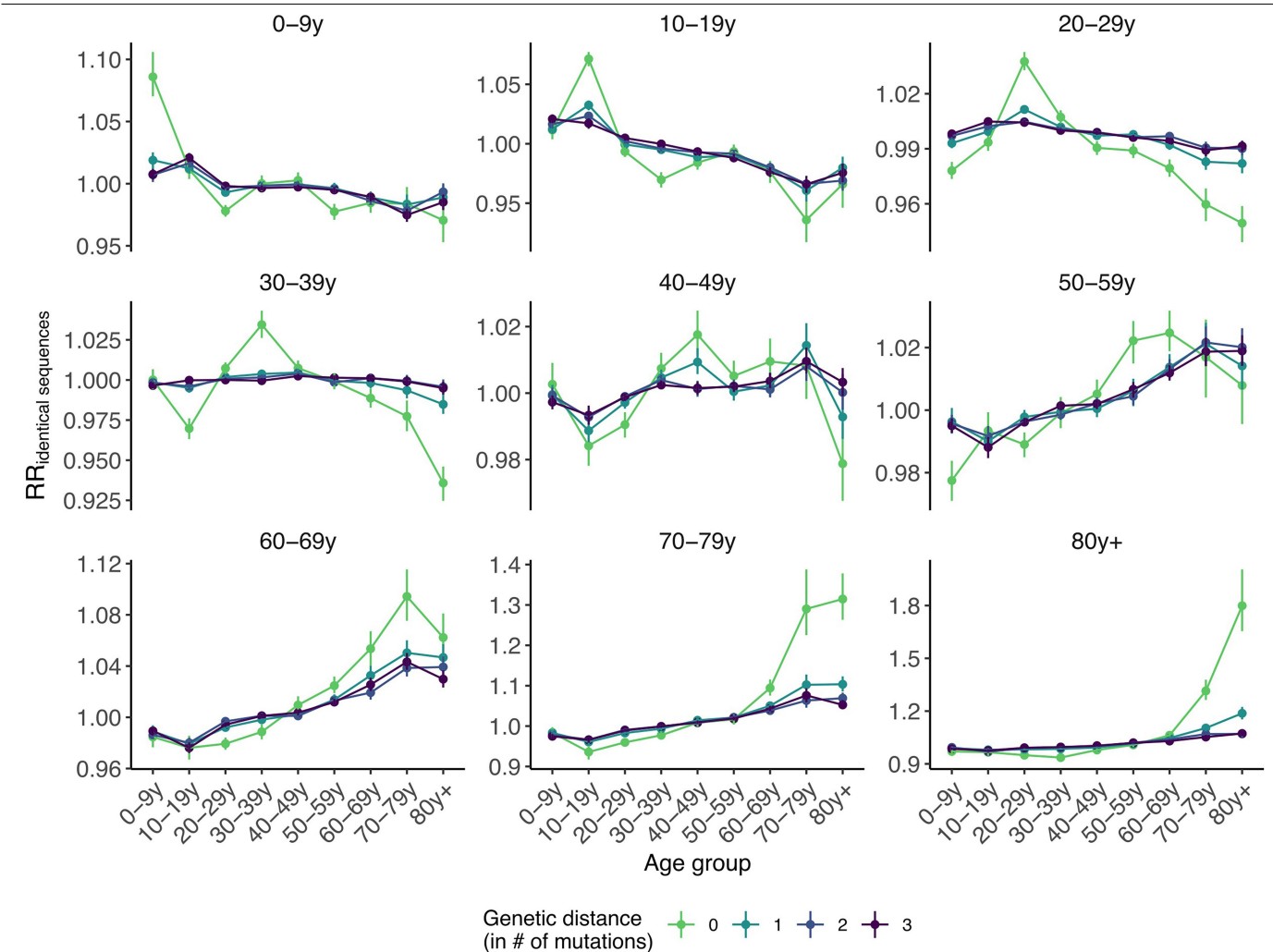

**Extended Data Fig. 6 | Relative risk for pairs sequences of being observed between two age groups depending on their genetic distance.** Vertical segments correspond to 95% confidence intervals obtained through subsampling.

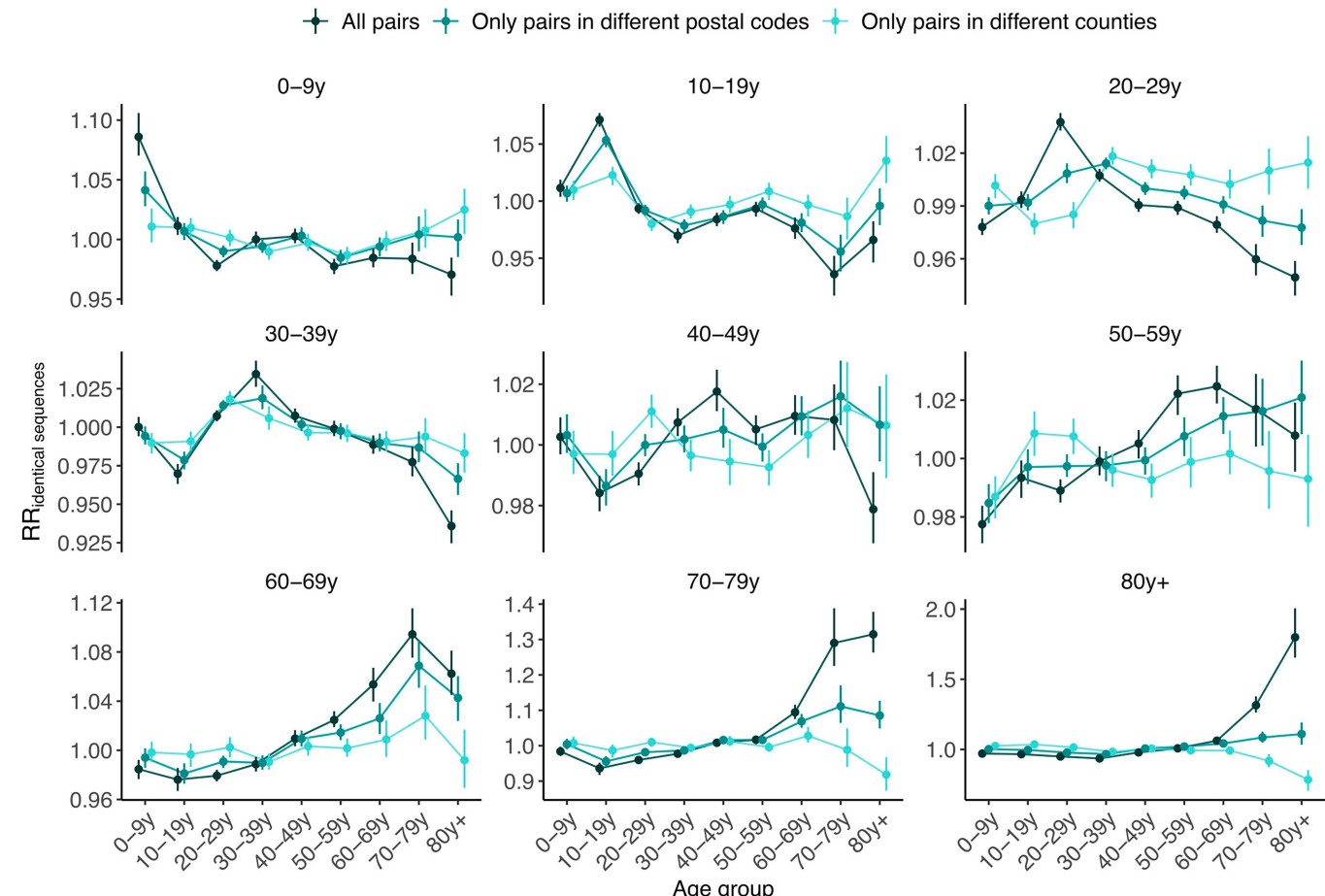

**Extended Data Fig. 7 | Impact of the spatial scale on the relative risk for pairs sequences of being observed between two age groups.** Vertical segments correspond to 95% confidence intervals obtained through subsampling.

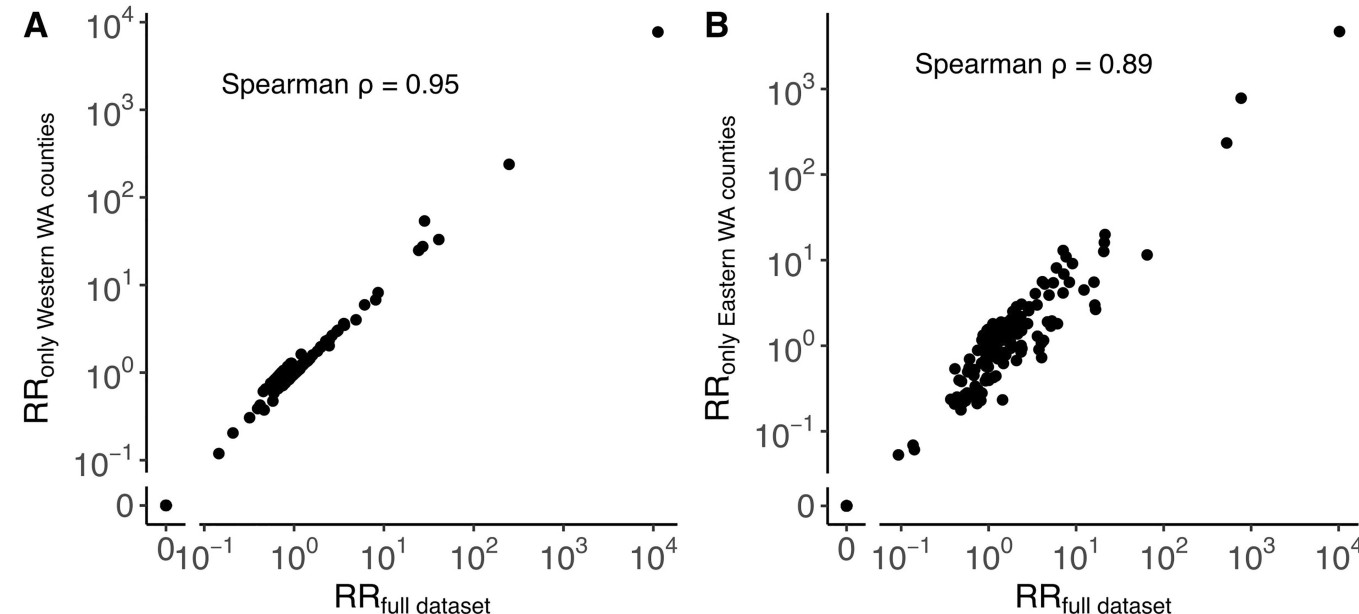

**Extended Data Fig. 8 | Impact of non-sampled locations on the computation of the RR. A**. Comparison between the relative risk of observing identical sequences between Western WA counties using only sequence in Western WA counties or the entire sequence dataset. **B**. Comparison between the relative risk of observing identical sequences between Eastern WA counties using only sequence in Eastern WA counties or the entire sequence dataset.

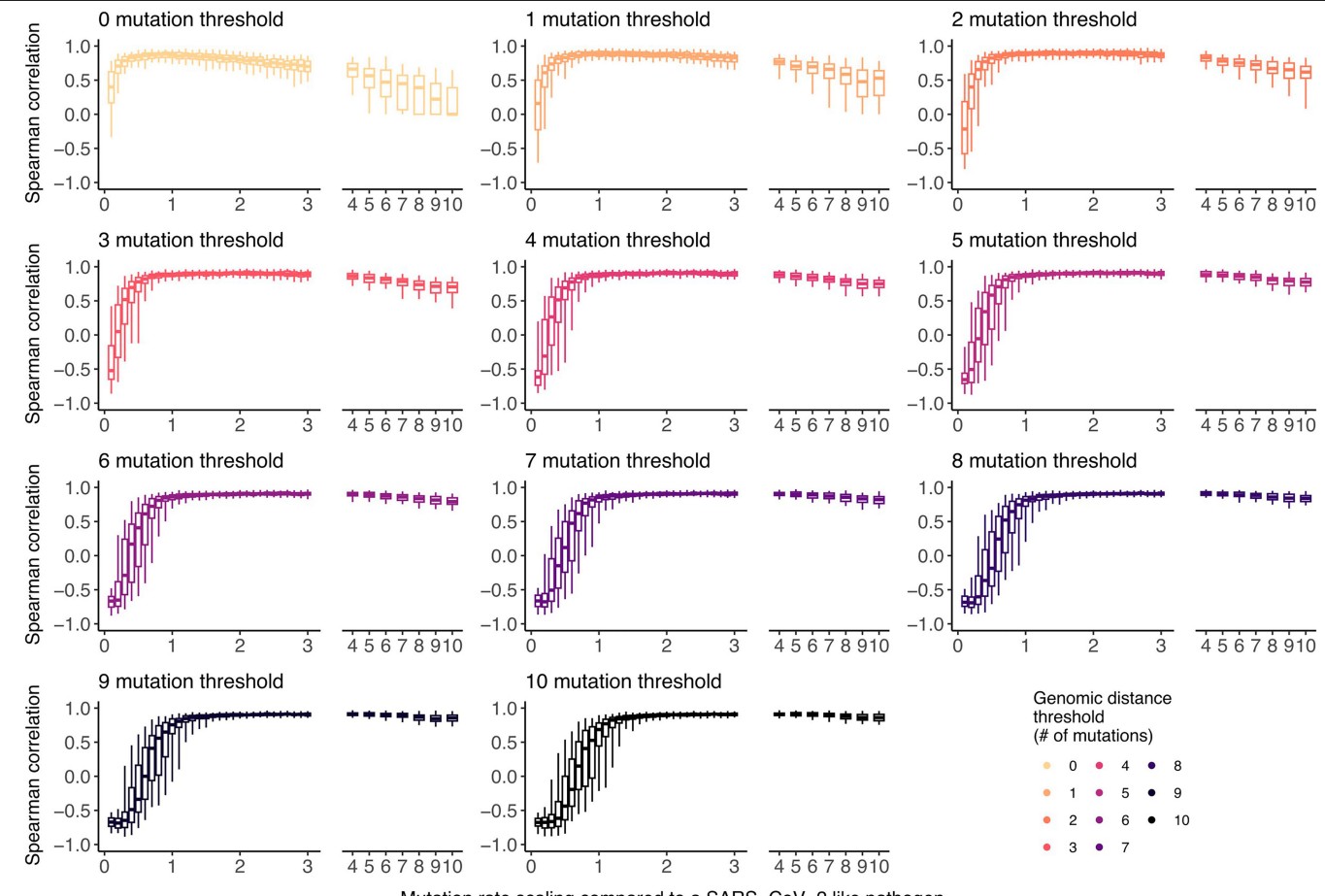

**Extended Data Fig. 9 | Impact of the pathogen's mutation rate on the optimal Hamming distance threshold to apply our RR framework.** Boxplots indicate Spearman correlation coefficients between the relative risk of pairs of sequences below the genomic distance threshold of being observed in two regions and the daily migration probability between these two regions. Boxplots indicate the 2.5%, 25%, 50%, 75% and 97.5% percentiles. See Methods for a description of the simulation approach.

**Extended Data Table 1 | Comparison between the relative risk of observing identical sequences between two geographic regions and the risk of movement between different geographies**

|  | At the county level | At the region level |
|---|---|---|
| *Spearman correlation $\rho$* |  |  |
| Mobile phone mobility | 35% | 61% |
| Workflow mobility data | 40% | 59% |
| Geographic distance | -35% | -48% |
| *Spearman correlation $\rho$ (without 0)* |  |  |
| Mobile phone mobility | 43% | 61% |
| Workflow mobility data | 56% | 59% |
| Geographic distance | -36% | -48% |
| *Variance explained (GAM)* |  |  |
| Mobile phone mobility | 60% | 81% |
| Workflow mobility data | 70% | 79% |
| Geographic distance | 32% | 57% |

We consider three data sources to inform the relative risk of movement between geographies: the relative risk for a visit to occur between two geographies (from mobile phone data), the relative risk for a work commute to occur between two geographies (from workflow data) and the geographic distance between geographies' centroids.

# Reporting Summary

## Statistics

For all statistical analyses, confirm that the following items are present in the figure legend, table legend, main text, or Methods section.

| n/a | Confirmed | |
|---|---|---|
| ☒ | ☐ | The exact sample size (*n*) for each experimental group/condition, given as a discrete number and unit of measurement |
| ☐ | ☒ | A statement on whether measurements were taken from distinct samples or whether the same sample was measured repeatedly |
| ☐ | ☒ | The statistical test(s) used AND whether they are one- or two-sided *Only common tests should be described solely by name; describe more complex techniques in the Methods section.* |
| ☐ | ☒ | A description of all covariates tested |
| ☒ | ☐ | A description of any assumptions or corrections, such as tests of normality and adjustment for multiple comparisons |
| ☐ | ☒ | A full description of the statistical parameters including central tendency (e.g. means) or other basic estimates (e.g. regression coefficient) AND variation (e.g. standard deviation) or associated estimates of uncertainty (e.g. confidence intervals) |
| ☐ | ☒ | For null hypothesis testing, the test statistic (e.g. *F*, *t*, *r*) with confidence intervals, effect sizes, degrees of freedom and *P* value noted *Give P values as exact values whenever suitable.* |
| ☒ | ☐ | For Bayesian analysis, information on the choice of priors and Markov chain Monte Carlo settings |
| ☒ | ☐ | For hierarchical and complex designs, identification of the appropriate level for tests and full reporting of outcomes |
| ☐ | ☒ | Estimates of effect sizes (e.g. Cohen's *d*, Pearson's *r*), indicating how they were calculated |

*Our web collection on statistics for biologists contains articles on many of the points above.*

## Software and code

Policy information about availability of computer code

| Data collection | Sequence data were downloaded from the GISAID EpiCov using the Nextstrain ncov-ingest pipeline. These sequences are publicly available with standard metadata (generally consisting of date of sample collection and sometimes county of sample collection). More detailed metadata curated by Washington State Department of Health (WA DOH) of county, postal code, age group and vaccination status were shared with the Fred Hutchinson Cancer Center. |
|---|---|
| Data analysis | Data were analysed using custom R scripts available at: https://github.com/blab/phylo-kernel-public/. The scripts were also deposited on Zenodo at the following DOI: https://doi.org/10.5281/zenodo.14398514 |

For manuscripts utilizing custom algorithms or software that are central to the research but not yet described in published literature, software must be made available to editors and reviewers. We strongly encourage code deposition in a community repository (e.g. GitHub). See the Nature Portfolio guidelines for submitting code & software for further information.

# Data

Policy information about availability of data

All manuscripts must include a data availability statement. This statement should provide the following information, where applicable:

- Accession codes, unique identifiers, or web links for publicly available datasets
- A description of any restrictions on data availability
- For clinical datasets or third party data, please ensure that the statement adheres to our policy

All sequences referenced in this manuscript are publicly shared to GISAID and are publicly available with standard metadata (generally consisting of date of sample collection and sometimes county of sample collection).
More detailed metadata curated by Washington State Department of Health (WA DOH) of county, postal code, age group and vaccination status were shared with the Fred Hutchinson Cancer Center under a Data Sharing Agreement for Confidential Data with an associated IRB.
These more detailed metadata are not currently shared publicly while we seek clearance with WA DOH.
GISAID accessions and a sequence-level acknowledgements table are provided in the GitHub repository associated with this manuscript. They were also deposited on Zenodo at the following DOI: https://doi.org/10.5281/zenodo.14398514.

# Research involving human participants, their data, or biological material

Policy information about studies with human participants or human data. See also policy information about sex, gender (identity/presentation), and sexual orientation and race, ethnicity and racism.

| | |
|---|---|
| Reporting on sex and gender | Data are not analysed by sex and gender. |
| Reporting on race, ethnicity, or other socially relevant groupings | Data are not analysed by race or ethnicity. |
| Population characteristics | We included all SARS-CoV-2 sequences collected through WA state genomic sentinel surveillance between 1 March 2021 and 31 December 2022. The age of individuals was grouped as: 0-9y, 10-19y, 20-29y, 30-29y, 40-49y, 50-59y, 60-69y, 70-79y and 80y+. |
| Recruitment | Sequence data were collected as part of Washington State genomic sentinel surveillance. The implementation of this surveillance system is described in: 10.3201/eid2902.221482. |
| Ethics oversight | The Washington State and University of Washington Institutional Review Boards determined this project to be surveillance activity and exempt from review; the need for informed consent was waived through this determination.<br>Under Washington State IRB Exempt Determination 2020-102, symptom onset date, age group, residence county, residence postal code and vaccination history was provided by the Washington Department of Health from the Washington Disease Reporting System for individuals with linked sequenced SARS-CoV-2 samples from March 1, 2021 through December 31, 2022. Sequencing and analysis of samples from the Seattle Flu Study was approved by the Institutional Review Board (IRB) at the University of Washington (protocol STUDY00006181).  Sequencing of remnant clinical specimens at UW Virology Lab was approved by the University of Washington Institutional Review Board (protocol STUDY00000408). |

Note that full information on the approval of the study protocol must also be provided in the manuscript.

# Field-specific reporting

Please select the one below that is the best fit for your research. If you are not sure, read the appropriate sections before making your selection.

☒ Life sciences    ☐ Behavioural & social sciences    ☐ Ecological, evolutionary & environmental sciences

For a reference copy of the document with all sections, see nature.com/documents/nr-reporting-summary-flat.pdf

# Life sciences study design

All studies must disclose on these points even when the disclosure is negative.

| | |
|---|---|
| Sample size | We used all available sequences from WA sentinel surveillance between 1 March 2021 and 31 December 2022 with linked metadata (116,791 sequences). |
| Data exclusions | When multiple sequences were linked to a de-identified patient ID, we only included the earliest sequence collected (for each of the patient with multiple sequences). This was done to avoid including sequences from the same individual and comparing individual to themselves (as these sequences are very likely to be identical when sampled during the same time period). This resulted in 114,306 SARS-CoV-2 sequences with unique patient ID. |
| Replication | NA |
| Randomization | NA |

| Blinding | NA |
|---|---|

# Reporting for specific materials, systems and methods

We require information from authors about some types of materials, experimental systems and methods used in many studies. Here, indicate whether each material, system or method listed is relevant to your study. If you are not sure if a list item applies to your research, read the appropriate section before selecting a response.

## Materials & experimental systems

| n/a | Involved in the study |
|---|---|
| ☒ ☐ | Antibodies |
| ☒ ☐ | Eukaryotic cell lines |
| ☒ ☐ | Palaeontology and archaeology |
| ☒ ☐ | Animals and other organisms |
| ☒ ☐ | Clinical data |
| ☒ ☐ | Dual use research of concern |
| ☒ ☐ | Plants |

## Methods

| n/a | Involved in the study |
|---|---|
| ☒ ☐ | ChIP-seq |
| ☒ ☐ | Flow cytometry |
| ☒ ☐ | MRI-based neuroimaging |

## Plants

| Seed stocks | NA |
|---|---|

| Novel plant genotypes | NA |
|---|---|

| Authentication | NA |
|---|---|

