## [Peer Review File · Nature]

Fine-scale patterns of SARS-CoV-2 spread from identical pathogen sequences

Corresponding Author: Dr Cécile Tran-Kiem

Version 0:

Reviewer comments:

Referee #1

(Remarks to the Author)

Tran-Kiem et al. describe a new methodology to analyse large scale pathogen genomic data from SARS-CoV-2 and more specifically to understand transmission across space and time in an urban area in the USA.

The authors find that using a simple approach based on identical sequences yields surprisingly accurate results in reconstructing transmission relationships between counties and that they are related to distance and human mobility. The general findings that mobility and distance are important in driving epidemic dynamics has been demonstrated elsewhere. However, I believe in this work they are more used to illustrate that the proposed framework reconstructs known relationships between spatial infectious disease transmission and human mobility.

Can the authors speculate as to why they observe a non-linear relationship between mobility and RR? Might this point towards an underlying epidemic process where highly connected regions form superclusters of transmission while those that are weakly connected have fewer chances for epidemics to take off? I am unsure about the exact mechanism but would be interested to hear the authors perspective.

The mobility data itself is biased and it has repeatedly been shown that biases from mobile phone data can lead to biased inferences. For example, do the authors know the privacy cutoff?

For me the comparison between the DTA method and the RR method is unclear. The paper argues that the identical sequences are important in tracking transmission events; method reconstructing transmission chains might be more appropriate as a comparison.

In the section about applicability beyond SARS-CoV-2 the authors do not mention specific pathogens. For example, could these methods be applied to datasets with higher mutation rates? If so, would the cutoff change, as in not using identical sequences but those with fewer than X mutations?

Are the authors able to comment on the directionality of how virus was transmitted between counties especially as multiple seedings become possible when multiple counties are infected?

The authors comment that no data from outside the study region is needed. I found that surprising because there could be (in theory) a seeding location just across state lines that is seeding the observed outbreaks.

##

In summary I found this to be an interesting manuscript and a nice demonstration of a new method to analyse densely sampled pathogen genomic datasets of SARS-CoV-2 where detailed metadata is available.

The manuscript was well written and the figures looked great.

(Remarks on code availability)

I have not reviewed the code in detail or to reproduce the analysis. I am happy though to do that at a later stage should it be required.

Referee #2

(Remarks to the Author)

This study uses identical sequences within large genome-sequencing datasets describing outbreaks to identify putative transmission patterns within and between counties in a region of the US. This is an interesting and effective approach, if not entirely novel. The study is generally performed to a high standard, clearly written and well explained. The metric used to identify transmission events correlates well with other measures of transmission.

A simple approach is used in this manuscript, identifying cases of infection with identical consensus sequences (or sometimes sequences with no more than k differences in consensus sequences). This is a useful metric, but the basic idea has been applied elsewhere. In the identification of clusters of connected cases a k -nucleotide cutoff between sequences has commonly been used to identify plausibly linked cases of infection (e.g. 1 difference Arons, et al, NEJM, 2020; ≤ 2 in Paltansing et al., J Hosp Infec, 2021). Allowing one or two substitutions provides a more inclusive metric of potential linkage, at the cost of including more cases of indirect transmission, but the idea is basically the same. This study adds the use of a normalisation for the extent of data collection in each county, and studies data on the level of a state rather than a healthcare facility, but in essence the approach builds upon previously-trodden ground. In this sense the straightforward comparison made between a nucleotide-based cutoff and phylogeographic methods misrepresents the body of previous work in this area.

- The results on age-specific drivers of transmission and patterns of travel were interesting but unsurprising; perhaps more could be said here. For example, mobility data correlates with patterns of transmission (e.g. McCrone et al, 2022), as people carry viruses with them as they move. Further, genomic similarity correlates with patterns of transmission due to the underlying dynamics of virus evolution. Is the intended point of this study that mobility data and age-specific contact data are made redundant by the information contained with genome sequences? If so, this could represent a substantial cost saving, or at the least, a refocusing of resources, in responses to outbreaks.

Methodology:

- The direction of transmission was estimated based upon times of sequence collection, but this is complex. For example, individuals in prison might be subject to a different testing regime to those outside. If symptom onset times were used to estimate directionality, the variation in the time from infection to symptom onset between individuals (e.g. He et al., Nature Medicine 2020) would make this a somewhat unreliable indicator of the directionality of transmission. If sequence collection was carried out on a non-systematic basis (e.g. anything other than a fixed number of days after symptom onset), the data would be more problematic.

- In so far as the statistic RR depends upon the epidemiology of the virus, the outputs from the approach differ between SARS-CoV-2 variants (e.g. Figure S2A). Are there ways in which strain-specific epidemiology could be normalised out, or accounted for in some way. The concern here is in relating the outputs of the method into actionable information: If the results of the method reflect changes in epidemiology more than anything else, does that make them quite confusing when presented in a public health context?

(Remarks on code availability)

- The authors supply code in the form of a series of R scripts. These perform the initial analysis of the sequence data, and then appear to generate many (but probably not all) of the figure panels.

- The instructions indicate how to use initial analysis script, and show how to apply it to a small synthetic dataset. This code is commented and seems fine, but no tests are provided.

- The remainder of the code does not have clear usage instructions, and generally does not appear to work without some of the output files, which are presumably not supplied because of the restrictions on the underlying data. It is generally possible to work out which script generates which panel, but these are not clearly labelled. There are only a few comments in these scripts.

- I recommend that, if it is impossible to supply the processed data files, that the authors further supply a new synthetic dataset to check that the whole pipeline works. I also recommend fully documenting the series of commands required to perform all of the analyses in the paper using this synthetic data, indicating clearly which figure panels are generated by which command.

Referee #3

(Remarks to the Author)

In this article, the authors present a novel approach to uncover the local scale epidemic dynamics by combining pathogen genomic data with contact age and residence data. This manuscript is an important contribution demonstrating how genomic data could be used in outbreak investigations. The presentation of the relative risk, a simple-to-interpret statistic, describes how the number of pairs of sequences observed in age or spatial subgroups differs from an expectation. The use of identical pairs to identify clusters is novel.

I do appreciate that the phylogenetic classification was used to ensure comparisons were made among viruses belonging to the same clade. Presumably, increasing the genetic distance increases the cluster sizes, although ensuring the phylogenetic relatedness becomes more challenging. While these viruses may belong to the same clade, they might not be a linked cluster. Was this accounted for? I don't doubt that this is a non-trivial question. But I wonder if including clusters with more divergent viruses might be impacting RR. One thing that is unclear to me is the size or number of clusters identified during this period under investigation. For example, would 12 identical pairs identified over the entire delta wave be equivalent to 12 identical sequences identified in the first week of the Delta wave? The first might indicate 12 separate household transmissions, but the latter might indicate a large-scale transmission event. Furthermore - I don't know if this additional cluster information would impact the results.

Finally - I appreciated the "Caveat," "Applicability beyond SARS-CoV-2," and "Perspective" sections.

Pg 12 typo in Caveats section. don't

Pg 13 Typo "pathogen's" could be deleted.

Pg 23 repetitive sentences

"This directly (sp?) input tree is not possible in real-world scenarios where the genealogical tree must be (noisily) estimated from empirical sequence data."

"In practice, the genealogical tree has to be estimated from empirical sequence data."

JBahl

(Remarks on code availability)

I did not download or execute any of the code. But, the code is well documented and dependencies all seem to be in order. All data required to reproduce to work is also in place.

Version 1:

Reviewer comments:

Referee #1

(Remarks to the Author)

The authors have performed additional sensitivity analyses, added a figure panel, and done a great job addressing the criticisms raised.

It would be great to see if this method will be applicable across other settings where sampling is less dense and for other pathogens.

(Remarks on code availability)

I have not used the code to perform the analysis.

Referee #2

(Remarks to the Author)

This manuscript has been improved by the authors in response to the review comments. I am now happy that the work is better presented in the context of existing research, and that the authors have given good answers to questions of methodology and code availability, either by providing helpful new material, or by clarifying limitations of the method.

Considering the impact that this method is likely to have, there are two justifications given in the introduction, which I would agree are the critical points of interest. Firstly, where sample data is collected in a biased manner, this method avoids the distortions that can arise with some other approaches to data. Secondly, this approach is computationally fast, with the key computational step being the calculation of distances between pairs of aligned genome sequences. My question is on this second point: Do we really need methods for dealing with millions of genomes?

I think in some fields at least this claim is slightly contentious. During the recent pandemic millions of viral genomes were collected, but there is some questioning around whether such efforts would or should be repeated another time around. Work by UKHSA during 2021 showed that sequencing 1000 community cases per week plus hospital cases was sufficient to detect novel variants in good time, and community sequencing is now fairly sparse. While the cost of sequencing was small compared to e.g. the overall cost of testing, I don't think it is clear that sequencing projects on the scale of SARS-CoV-2 are seen as an important component in future public health efforts.

To support the claim that methods which analyse millions of genomes are of critical importance, could the authors say a little more on the information that can be discerned by having very large numbers of genomes, versus say, a few thousand? This is touched upon in the discussion, but given its role in the justification of the method, it seems that more could be said. Do millions of genomes tell us what thousands don't?

(Remarks on code availability)

A postdoc in my group carried out the code review first time around, but has now moved on to a new position and was not available to do a re-review. I have read through the comments made in response to the review, and had a look at the code without conducting a full code review. I am reasonably satisfied that the authors have responded satisfactorily to the comments made, somewhat leaving the matter on trust.

Referee #3

(Remarks to the Author)

The detailed and thorough response has satisfied all of my concerns. The entire response document provides nice detail and examination of the study.
